# Endogenous nicotinamide riboside metabolism protects against diet-induced liver damage

Audrey Sambeat[1,3], Joanna Ratajczak[1,2,3], Magali Joffraud[1], José L. Sanchez-Garcia[1], Maria P. Giner[1], Armand Valsesia [1], Judith Giroud-Gerbetant[1], Miriam Valera-Alberni[1,2], Angelique Cercillieux[1,2], Marie Boutant[1], Sameer S. Kulkarni[1], Sofia Moco [1] & Carles Canto [1,2]

Supplementation with the $NAD^+$ precursor nicotinamide riboside (NR) ameliorates and prevents a broad array of metabolic and aging disorders in mice. However, little is known about the physiological role of endogenous NR metabolism. We have previously shown that NR kinase 1 (NRK1) is rate-limiting and essential for NR-induced $NAD^+$ synthesis in hepatic cells. To understand the relevance of hepatic NR metabolism, we generated whole body and liver-specific NRK1 knockout mice. Here, we show that NRK1 deficiency leads to decreased gluconeogenic potential and impaired mitochondrial function. Upon high-fat feeding, NRK1 deficient mice develop glucose intolerance, insulin resistance and hepatosteatosis. Furthermore, they are more susceptible to diet-induced liver DNA damage, due to compromised PARP1 activity. Our results demonstrate that endogenous NR metabolism is critical to sustain hepatic $NAD^+$ levels and hinder diet-induced metabolic damage, highlighting the relevance of NRK1 as a therapeutic target for metabolic disorders.

[1] Nestlé Research, EPFL Innovation Park, 1015 Lausanne, Switzerland. [2] School of Life Sciences, Ecole Polytechnique Fédérale de Lausanne (EPFL), 1015 Lausanne, Switzerland. [3] These authors contributed equally: Audrey Sambeat, Joanna Ratajczak. Correspondence and requests for materials should be addressed to C.C. (email: carlos.cantoalvarez@rd.nestle.com)

Nicotinamide riboside (NR) is a natural precursor for nicotinamide adenine dinucleotide (NAD$^+$) biosynthesis[1]. NAD$^+$ levels play a crucial role in energy homeostasis and genome stability by acting as a substrate for different families of enzymes, such as sirtuins (SIRTs) and poly(ADP-ribose) polymerases (PARPs)[2,3]. NAD$^+$ bioavailability declines in situations of metabolic disease, including diet-induced obesity and non-alcoholic fatty liver disease (NAFLD)[4], as well as during aging, fostering age-related physiological decline[5–9].

Restoring or enhancing NAD$^+$ levels through NR or nicotinamide mononucleotide (NMN) supplementation has been shown to prevent/treat a broad range of metabolic and age-related disorders[3]. For example, dietary supplementation with NR prevented diet-induced obesity and glucose intolerance in mice[10,11]. NR also had remarkable effects on hepatic metabolism, conferring resistance to hepatic lipid accumulation and reversing NAFLD states[4,11]. At the molecular level, the benefits of NR/NMN supplementation have been associated with the prevention of mitochondrial dysfunction at least in part via the activation of sirtuin enzymes[4,8]. Upon high-fat diet (HFD) feeding, however, the activity of another major NAD$^+$-consuming enzyme, PARP1, is induced in order to repair DNA damage, thereby competing with sirtuins for intracellular NAD$^+$ resources[12]. Thus, the maintenance of NAD$^+$ levels is essential to sustain PARP and sirtuin activities in situations of metabolic stress and delineates the basis for the therapeutic benefits of exogenous NR supplementation.

NR drives NAD$^+$ biosynthesis through a highly conserved path from yeast to mammals. It is initiated with the phosphorylation of NR by NR Kinases (NRK1 or NRK2), generating NMN which is subsequently converted in NAD$^+$ via NMN adenylyl transferase enzymes (NMNATs)[1]. Despite its ubiquitous presence, NRK1 is highly expressed in the liver, whereas the NRK2 protein can only be detected in heart and skeletal muscle[13]. Taken together, these observations suggest that endogenous NR utilization and NRK1 activity could be essential to sustain hepatic metabolic functions in non-supplemented conditions. Nevertheless, the physiological relevance of NR metabolism and whether its utilization provides any biological advantage compared to other NAD$^+$ precursors are not understood.

In order to investigate the native – i.e., in the absence of supplementation – role of NR metabolism on the onset of metabolic diseases, we used whole-body NRK1 knock out mice (NRK1 KO), which are unable to utilize intact NR as an NAD$^+$ biosynthetic substrate in most tissues. NRK1 KO mice display reduced hepatic glucose production capacity and mitochondrial respiration. These defects are corroborated in liver-specific NRK1 KO mice (NRK1 LKO), thereby proving the tissue autonomous effect of NRK1 deletion in the liver. When NRK1 LKO mice are submitted to a HFD, the development of steatosis and hepatic insulin resistance are strikingly aggravated due to impaired mitochondrial fatty acid β-oxidation (FAO). Furthermore, at the molecular level, liver-specific NRK1 deletion induces NAD$^+$ depletion leading to decreased PARP1 activity, thus exacerbating HFD-induced hepatic DNA damage. Finally, we show that these defects cannot be reversed by nicotinamide (NAM) supplementation. Altogether, our results demonstrate that NR is a critical endogenous substrate to maintain NAD$^+$ levels and hepatic oxidative metabolism in situations of metabolic stress and DNA damage. Further, we demonstrate that NR deficiency cannot be compensated by NAM, arguing that NAD$^+$ precursors are not always exchangeable.

## Results
### NRK1 deficiency impairs hepatic gluconeogenic capacity.
NRK1 is essential and rate-limiting for NAD$^+$ synthesis from exogenous NR and NMN in primary hepatocytes[13]. Furthermore,

NRK1 is highly expressed in mouse liver[13]. Therefore, we aimed to investigate the effect of NRK1 deletion – i.e., the inability to utilize NR for NAD$^+$ synthesis – on metabolic homeostasis and hepatic function in vivo. Given the critical role of the liver in glycemic control, we subjected NRK1 KO mice to intraperitoneal glucose, insulin and pyruvate tolerance tests. NRK1 KO mice exhibited similar glucose excursion curves to wild-type (WT) mice when challenged with glucose or insulin (Supplementary Fig. 1a, b). However, when injected with pyruvate, the glycemia in NRK1 KO mice rose to a lesser degree compared to control mice as shown by the difference in AUC between the two genotypes (Fig. 1a), suggesting an impaired gluconeogenic capacity. While basal glycemia after an overnight fast (Fig. 1a) or 24-h fasting (Fig. 1b) was similar between genotypes, we noticed that NRK1 KO mice exhibited lower glucose levels after 6 h of fasting, further supporting the concept of defective gluconeogenic capacity in NRK1 KO mice. The protein levels of the gluconeogenic enzyme phosphoenolpyruvate carboxykinase (PEPCK) and the mRNA expression of the PPARγ coactivator 1α (*Pgc1*α), a key driver of the gluconeogenic gene program[14,15], were similar between WT and NRK1 KO mice during the fasting time course (Fig. 1c, d). However, glucose-6-phosphatase (*G6Pase*) mRNA levels were lower in NRK1 KO mice at early fasting stages (Fig. 1d).

The initial steps of pyruvate driven gluconeogenesis occur in the mitochondria where pyruvate carboxylase (PCB) converts pyruvate into oxaloacetate. In contrast, glycerol enters the gluconeogenic pathway in the cytosol, where it is converted to glycerol 3-phosphate (G3P) via glycerol kinase (GK). Hence, we next used glycerol as the gluconeogenic precursor and observed similar glucose excursion curves between genotypes (Fig. 1e), indicating that the impaired glucose production from pyruvate could result from mitochondrial defects. Furthermore, it also suggests that the decrease in *G6Pase* expression we detected is not the limiting factor for defective glucose production from pyruvate in NRK1 KO mice. In this sense, we did not detect any change in the protein level of PCB (Supplementary Fig. 1c), suggesting that other aspects of mitochondrial function may be accountable for the defects observed. Thus, we examined mitochondrial respiration by high-resolution respirometry analysis. NRK1 KO mice manifested marked defects in mitochondrial respiratory capacity both in the fed or fasted state (Fig. 1f). Mitochondrial respiration through Complex I + II and maximal electron transport system (ETS) capacity were lower in the liver preparations from NRK1 KO mice. In contrast, we did not detect differences in skeletal muscle respiratory capacity (Supplementary Fig. 1d), most likely due to the relatively low protein levels of NRK1 in this tissue[13]. Taken together, these results validate the role of NRK1 and endogenous NR metabolism in the maintenance of hepatic mitochondrial function, in turn affecting gluconeogenic capacity.

### NRK1 deletion aggravates hepatic insulin-resistance.
To address whether the hepatic phenotype of NRK1 KO mice is tissue autonomous, we generated NRK1 liver-specific knockout mice (NRK1 LKO) by crossing NRK1$^{loxP/loxP}$ mice with mice expressing Cre recombinase under the albumin promoter. This led to a complete ablation of NRK1 expression in the liver (Fig. 2a), without affecting NRK1 levels in other tissues (Supplementary Fig. 2a). NRK1 deficiency was neither compensated by NRK2, which was undetectable in the liver, nor by changes in NAMPT protein levels (Fig. 2a).

When mice were fed with a low-fat diet, the body weight and composition of NRK1 LKO mice were indistinguishable from control littermates (Supplementary Fig. 2b). Glucose excursion was similar between control and NRK1 LKO mice in response to a glucose or insulin challenge (Supplementary Fig. 2c, d).

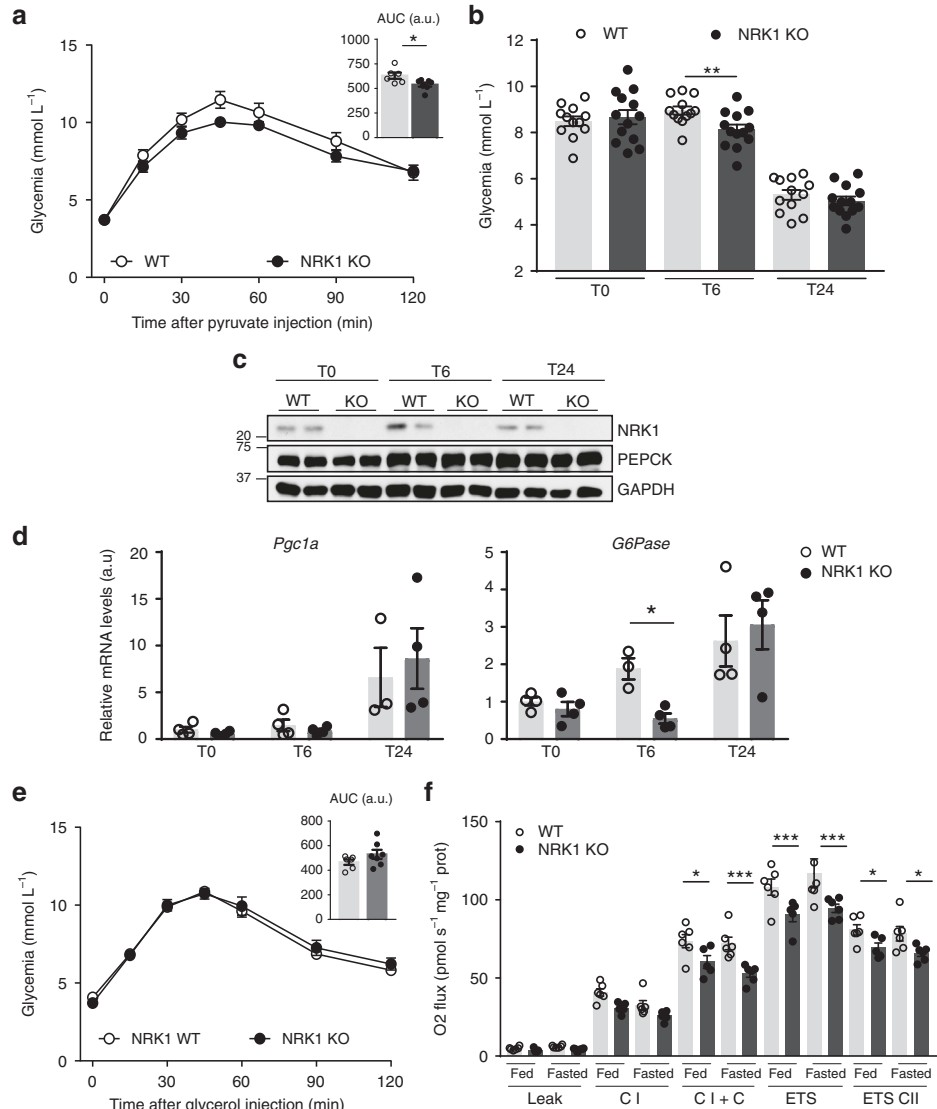

**Fig. 1** NRK1 KO mice display lower gluconeogenic capacity. **a** Blood glucose levels during intraperitoneal pyruvate tolerance tests in NRK1 KO ($n = 7$) and control (WT) ($n = 6$) mice. **b** Blood glucose levels in NRK1 KO ($n = 13$) and WT ($n = 12$) mice subjected to a fasting time course. **c** Livers were snap frozen at the indicated time-points during a fasting time course. A piece of liver was then homogenized for protein extraction. Then, 20 μg of protein were used to evaluate NRK1, PEPCK, and GAPDH protein levels through western blot. **d** Livers were snap frozen at the indicated time points during fasting time course. Then, RNA was extracted to evaluate mRNA levels of genes involved in gluconeogenesis ($n = 4$ mice per group). **e** Blood glucose level during intraperitoneal glycerol tolerance test ($n = 6$ for WT mice, $n = 7$ for NRK1 KO mice). **f** Mitochondrial respiratory capacity properties were evaluated in liver homogenates from NRK1 KO and WT mice in fed and 24h-fasted conditions ($n = 6$ for WT mice; $n = 5$ for NRK1 KO mice). Results shown are mean ± SEM; * and *** indicate statistical difference between genotypes at $p < 0.05$ and $p < 0.001$, respectively. The individual values and statistical tests used for each panel can be found in the Source Data file

However, as in the whole body NRK1 KO mice, NRK1 LKO mice displayed impaired glucose production after a pyruvate bolus (Supplementary Fig. 2e), even if no differences were observed in the expression of gluconeogenic-related genes such as *Pepck* or *G6Pase*, or the transcriptional regulator *Pgc1α* (Supplementary Fig. 2f). These observations indicate that the impaired response to pyruvate observed in NRK1 KO mice is liver autonomous. Furthermore, it confirms that the reduced *G6Pase* expression observed in whole body NRK1 KO mice is not at the root of the gluconeogenic defects.

We next explored the response of NRK1 LKO mice to diet-induced metabolic damage by placing them on a high fat diet (HFD). Body weight gain and composition were comparable between control and NRK1 LKO mice (Fig. 2b), as well as food intake and daily activity (Fig. 2c, d). While, as expected, chronic HFD exposure increased glycemia in control mice fasted for 6 h, this increase was largely prevented in NRK1 LKO animals (Fig. 2e). NRK1 LKO mice on HFD developed glucose intolerance (Fig. 2f) and insulin resistance (Fig. 2g) as demonstrated by increased AUC and a reduced AAC, respectively. The altered glucose profile in NRK1 LKO mice during the ipGTT did not come in parallel to alterations of gluconeogenic gene expression (Supplementary Fig. 2g) nor variations in insulinemia, which were comparable between genotypes during the test (Supplementary Fig. 2h). Further testifying for hepatic insulin resistance, insulin-induced Akt phosphorylation was compromised in the livers of high-fat fed NRK1 LKO mice (Fig. 2h). In order to rule out the potential contribution of systemic factors, we performed a similar experiment in primary hepatocytes exposed to a lipid rich medium containing 0.2 mM oleic acid and 0.2 mM palmitate to

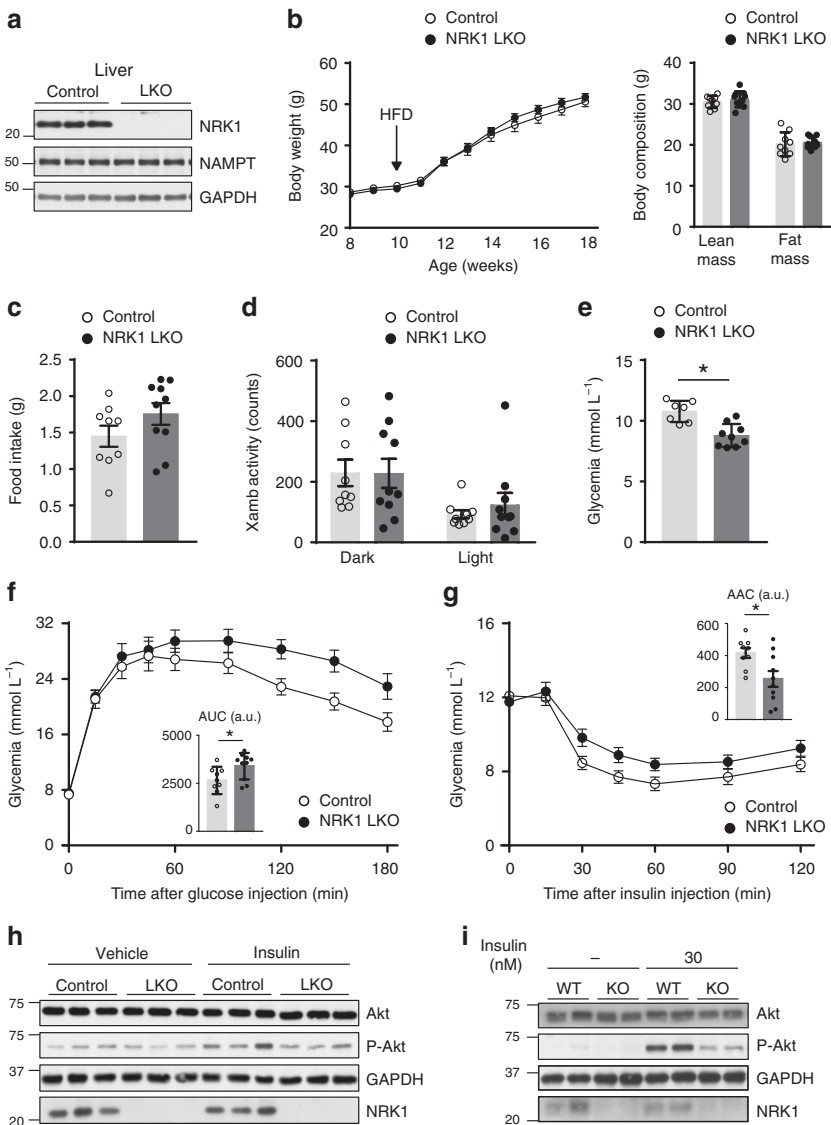

**Fig. 2** Liver-specific NRK1 deficiency exacerbates diet-induced insulin resistance. **a** Validation of NRK1 deletion and protein level of NAMPT and GAPDH in liver homogenates from NRK1 LKO and control (Ctrl) mice. **b** Body weight and body composition of NRK1 LKO ($n = 10$) and control ($n = 9$) mice on high-fat diet (HFD) at the indicated ages. **c, d** Food intake **c** and locomotor activity **d** of NRK1 LKO ($n = 9$) and control ($n = 8$) mice on HFD at 18 weeks of age. **e** Glycemia levels after 6h-fasting in NRK1 LKO ($n = 9$) and control ($n = 7$) mice on HFD at 24 weeks of age. **f, g** Blood glucose level during intraperitoneal glucose ($n = 8$ mice in the control group, $n = 10$ mice in the NRK1 LKO group) **f** and insulin **g** tolerance test in NRK1 LKO and control mice on HFD ($n = 9$ mice for the control group, $n = 10$ mice for the NRK1 LKO group) at 20 and 22 weeks of age, respectively. **h** Protein level of Akt, P-Akt, GAPDH, and NRK1 in the liver of NRK1 LKO and Ctrl mice on HFD 15 min after insulin injection (1U kg$^{-1}$) ($n = 3$ mice per group). **i** Protein levels of Akt, P-Akt, GAPDH and NRK1 in primary hepatocytes from NRK1 KO and WT mice stimulated with insulin at the indicated concentration. Results shown are mean ± SEM, * indicates statistical difference vs the respective control value at $p < 0.05$. The individual values and statistical tests used for each panel can be found in the Source Data file

mimic dietary lipid overload. Remarkably, after insulin stimulation, Akt phosphorylation was lower in NRK1 null hepatocytes compared to those from WT mice (Fig. 2i). These observations argue for a dramatic exacerbation of diet-induced glucose intolerance and hepatic insulin resistance, thereby pointing towards a crucial role of NRK1 and endogenous NR to maintain hepatic function upon diet-induced metabolic damage.

**Hepatic NRK1 deletion impairs fatty acid oxidation capacity.**
The liver weight in NRK1 LKO mice on HFD was higher than in control animals, while no differences were observed on LFD (Fig. 3a). Although plasma cholesterol, triglycerides (TG) or free

fatty acids (FFA) levels did not show significant changes (Supplementary Fig. 3a), hepatic TG levels increased in NRK1 LKO mice on HFD (Fig. 3b) suggesting an increased hepatic lipid accumulation, which was corroborated by Oil Red-O staining (Fig. 3c).

We next aimed to understand the factors influencing hepatic steatosis in NRK1 LKO mice. The mRNA levels of lipogenic genes such as *Fas* or *Scd1* and related transcriptional regulators were similar between genotypes (Fig. 3d). Similarly, the levels of the AMP-activated protein kinase (AMPK), a key cellular controller of lipogenesis and lipid metabolism, and its phosphorylation state were similar between genotypes (Supplementary Fig. 3b). In contrast, we detected increased mRNA levels of *Cd36, Ldlr, Lrp*

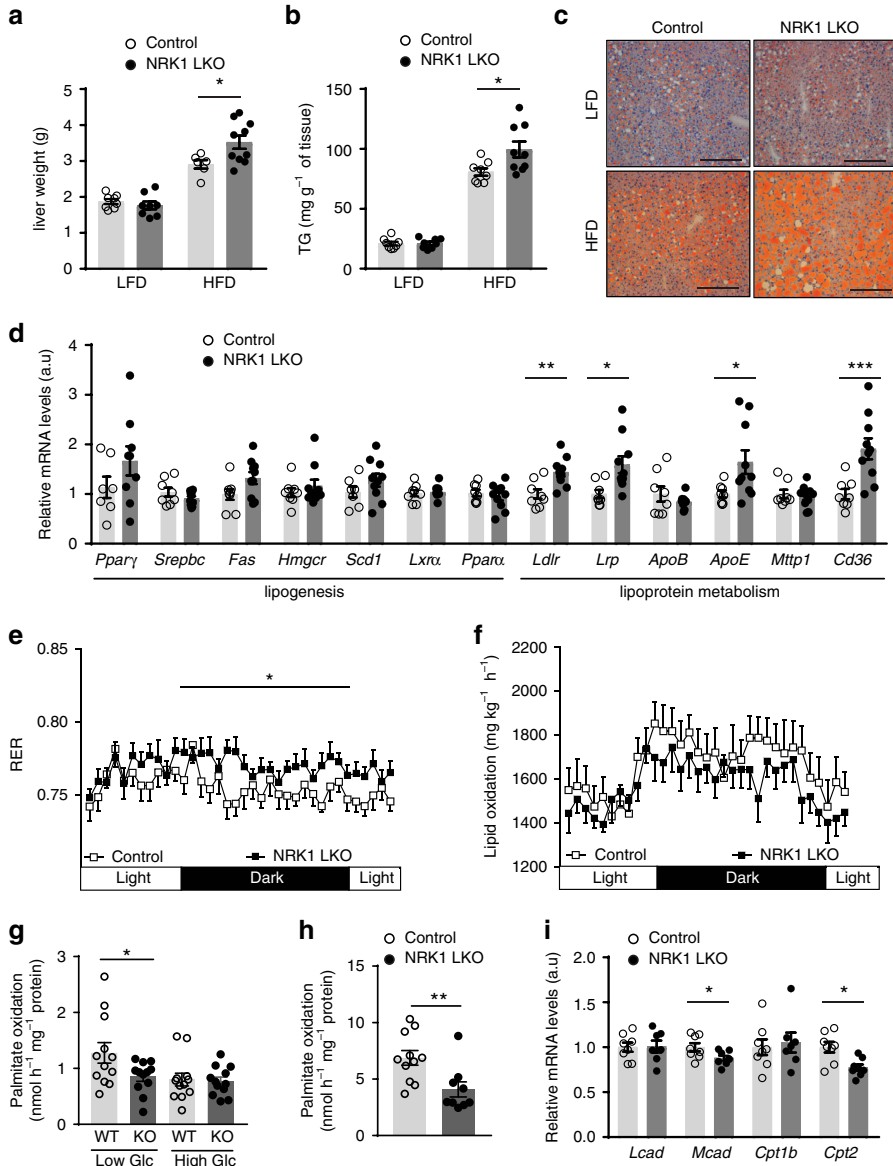

**Fig. 3** NRK1 deficiency promotes hepatic steatosis by impairing fatty acid oxidation. **a, b** Liver weight **a** and triglycerides (TG) levels **b** in liver of NRK1 LKO and control mice on LFD and HFD ($n = 7$ mice for the control group; $n = 10$ mice for the NRK1 LKO group; 24 weeks of age). **c** Oil-Red O staining of liver sections from NRK1 LKO and control mice on LFD and HFD (scale bars: 200 μm). **d** mRNA level of genes involved in lipogenesis and lipoprotein metabolism in the liver of HFD-fed NRK1 LKO and control mice ($n = 8$ mice for the control group; $n = 10$ mice in the NRK1LKO group). **e** Respiratory exchange ratio (RER) in NRK1 LKO and control mice on HFD ($n = 9$ mice for the control group; $n = 10$ mice for the NRK1LKO group). **f** Lipid oxidation of NRK1 LKO and control mice on HFD. Data calculated from indirect calorimetry measurements: lipid oxidation = $(1.67 \times VO_2) - (1.67 \times VCO_2)$[43] ($n = 9$ mice for the control group; $n = 10$ mice for the NRK1 LKO group). **g** Fatty acid oxidation in primary hepatocytes isolated from 10 to 20 week-old WT and NRK1 KO mice. [3]H-Palmitate oxidation was measured in conditions of low and high glucose ($n = 12$ for all groups). **h** Fatty acid oxidation in primary hepatocytes isolated from 8 to 12 week-old WT and NRK1 KO mice. [3]H-Palmitate oxidation was measured in condition of lipid overload. **i**. mRNA level of genes involved in β-oxidation in the liver of HFD-fed NRK1 LKO and control mice ($n = 8$ for WT hepatocytes; $n = 7$ for KO hepatocytes). Results shown are mean ± SEM, *, **, and *** indicate statistical difference vs. respective control group at $p < 0.05$, $p < 0.01$ and $p < 0.001$, respectively. The individual values and statistical tests used for each panel can be found in the Source Data file

and *ApoE* (Fig. 3d), which suggest higher lipid incorporation from the circulation. This higher influx could prompt hepatic lipid accumulation if NRK1 LKO mice had impaired hepatic fatty acid catabolism. Supporting this possibility, indirect calorimetry studies indicated that, despite similar daily energy expenditure (Supplementary Fig. 3c), the respiratory exchange ratio (RER) was higher in HFD-fed NRK1 LKO mice during the dark phase (Fig. 3e). There was no change in whole body glucose oxidation between genotypes (Supplementary Fig. 3d), and lipid oxidation rates tended to be lower in NRK1 LKO compared to control

littermates, albeit not significantly (Fig. 3f). These changes, however, were detectable only when mice were challenged by HFD, as mice on LFD did not exhibit changes in RER or energy substrates oxidation (Supplementary Fig. 3e–g).

In order to further ascertain the critical impact of hepatic NRK1 on FAO capacity, we directly evaluated palmitate oxidation rates in primary hepatocytes. Hepatocytes lacking NRK1 showed no alterations in palmitate oxidation rates when incubated in regular high-glucose concentration (25 mM) medium but failed to increase lipid oxidation rates when they were shifted to a lower

glucose concentration medium (Fig. 3g). Similarly, when incubated with high lipid containing medium to mimic the HFD intervention, NRK1-null hepatocytes were unable to sustain comparable palmitate oxidation rates as WT hepatocytes (Fig. 3h). In line with these observations, we detected a lower expression of β-oxidation-related genes, including medium-chain specific acyl-Coenzyme A dehydrogenase (Mcad) and carnitine palmitoyl-transferase 2 (Cpt2) (Fig. 3i). Taken together, these data show that NRK1 deficient hepatocytes fail to properly increase lipid oxidation rates when metabolically challenged.

A critical element to sustain lipid oxidation is mitochondrial respiratory capacity. In line with the observations in whole body NRK1 knockout mice (Fig. 1f), mitochondrial respiratory capacity in liver homogenates from NRK1 LKO mice was impaired compared to control littermates (Supplementary Fig. 4a), and these defects were further accentuated by HFD feeding (Supplementary Fig. 4b). Similarly, mitochondrial respiratory capacity defects were also more pronounced in the whole body NRK1 KO mice (Supplementary Fig. 4c), supporting the idea that this is a genuine effect of NRK1 deletion. Strikingly, mitochondrial DNA content, citrate synthase activity and mitochondrial-related genes expression were similar between NRK1 LKO mice and control littermates (Supplementary Fig. 4d–f). In addition, the protein levels of respiratory complexes were similar between genotypes (Supplementary Fig. 4g, h). These results demonstrate that NRK1 deficiency leads to impaired mitochondrial respiratory capacity, which stems from intrinsic mitochondrial dysfunction rather than a lower mitochondrial content.

Collectively, we demonstrated that NRK1 LKO mice on LFD display only minor metabolic abnormalities. However, when prompted towards lipid utilization, by either HFD in vivo or by lipid overload in vitro, NRK1 ablated hepatocytes displayed an inability to oxidize lipids efficiently, promoting the development of insulin resistance and hepatic steatosis.

**Exacerbated diet-induced hepatic damage in NRK1 LKO mice.** Dysregulation of hepatic lipid metabolism has been largely associated with liver damage[16]. Accordingly, we detected significantly increased plasma levels of alanine transaminase (ALT) and aspartate transaminase (AST), two circulating markers of liver damage, in HFD-fed NRK1 LKO mice compared to control mice (Fig. 4a). Hematoxylin and eosin (H&E) staining in liver sections revealed increased infiltration of inflammatory cells in NRK1 deficient specimens (Fig. 4b top). This was also corroborated by a ~ 4-fold increase in CD45 positive cells, which is a leukocyte-specific marker (Fig. 4b bottom), as well as by an increased expression of inflammation markers such as Mcp1, Il-1β or Tnfα (Fig. 4c). Collectively, the exacerbated steatosis and insulin resistance along with the enhanced inflammatory profile suggest that NRK1 LKO progressed towards a more advanced stage of NAFLD. Solidifying this point, we observed a higher ratio of apoptotic cells by 20% and Ki-67 positive proliferating cells by 50% in the liver of NRK1 LKO mice on HFD (Fig. 4d). In addition, the fibrotic areas increased by 2.5-fold of in NRK1 LKO mice both in pericellular and perivascular structures (Fig. 4d), aligned with an increase in the mRNA levels of bona fide fibrosis markers including αSma and Pai-1 (Fig. 4c). Altogether, our results suggest that NRK1 deletion aggravated the effect of the HFD on the liver, promoting the progression from steatosis towards steatohepatitis characterized by liver steatosis, inflammation, and fibrosis.

**NRK1 deletion limits PARP activity and enhances DNA damage.** One of the reasons why overt physiological phenotypes were only observed in NRK1 LKO mice on HFD may come from

the fact that hepatic $NAD^+$ content was not affected in NRK1 LKO mice on LFD (Fig. 5a). However, when exposed to HFD, NRK1 LKO mice failed to sustain hepatic intracellular $NAD^+$ levels (Fig. 5a). This decrease can be fully attributed to a reduction in $NAD^+$ levels in the nucleo-cytoplasmic fraction, as mitochondrial $NAD^+$ levels were similar between genotypes (Supplementary Fig. 5a). We also detected lower $NAD^+$ content in isolated NRK1 null primary hepatocytes when subjected to lipid overload (Supplementary Fig. 5b). The decline of hepatic $NAD^+$ levels in HFD-fed NRK1 LKO mice occurred in parallel to an accumulation of NR, while NMN levels significantly decreased and NAM levels remained unaffected (Fig. 5b). Moreover, NAMPT mRNA and protein levels were comparable between HFD-fed control and NRK1 LKO mice (Fig. 5c, d). We also analyzed the hepatic mRNA level for multiple enzymes involved in the de novo and Preiss–Handler routes for $NAD^+$ biosynthesis which were unaffected by NRK1 deletion, with the exception of a 20% decline in Nmnat3 expression (Fig. 5c). Considering that even full deletion of Nmnat3 does not alter hepatic $NAD^+$ content[17], it seems likely that impaired NR utilization is at the root of the $NAD^+$ decline observed in HFD-fed NRK1 LKO mice. Therefore, NRK1 LKO mice on HFD are deficitary in $NAD^+$ levels even when NAM is available and NRK1-independent $NAD^+$ synthesis paths are unaltered.

It must be noted that HFD feeding did not influence hepatic $NAD^+$ levels in control mice (Fig. 5a). Nevertheless, total $NAD^+$ levels do not testify for $NAD^+$ turnover rates. Indeed, HFD feeding led to a marked increase in N-methyl-X-pyridone-5-carboxamide (MeXP, being Me2P or/and Me4P) in control mice (Supplementary Fig. 5c), one of the products of NAM metabolism, indicative of increased $NAD^+$-consuming enzymatic activity. This could be due, in part, to the increased expression level of the nicotinamide N-methyltransferase enzyme (Nnmt) which initiates the routing of NAM towards methylation and oxidation (Supplementary Fig. 5c). Of note, NRK1 protein levels are not affected by HFD in control mice (Supplementary Fig. 5d), suggesting that endogenous NRK1 levels can sustain the higher demand for $NAD^+$ biosynthesis generated by HFD in control mice. Altogether, these observations illustrate that in situations of high $NAD^+$ turnover the utilization of NR might be critical to sustain $NAD^+$ levels.

Sirtuins are a family of enzymes closely related to metabolic sensing and whose activity can be rate-limited by $NAD^+$, especially in the case of SIRT1 and SIRT3[2]. Interestingly, the hepatic deficiency of SIRT1 as well as the whole body deletion of SIRT3 have been linked to increased susceptibility to hepatic steatosis and related metabolic complications[18,19]. Hence, we evaluated the activity of different sirtuins, i.e.,: SIRT1, SIRT3 and SIRT5, by measuring acetylation levels of NF-κB and the profile of acetylation and malonylation of mitochondrial proteins, respectively. However, we did not observe differences in these protein modifications between genotypes (Supplementary Fig. 5e–g).

$NAD^+$ is also a mandatory co-substrate for poly(ADP-ribose) polymerases (PARPs), including PARP1, which is critical for DNA damage repair. Therefore, we asked whether reduction of intracellular $NAD^+$ levels in the liver of NRK1 LKO mice affects PARP1 activity. We assessed PARP1 protein level, as well as global poly(ADP-ribose) (PAR) levels, which reflect PARP1 PARylation activity. PARP1 and PAR levels were lower by about 50% and 25% in NRK1 LKO mice on LFD and HFD, respectively (Fig. 5d and Supplementary Fig. 5h). The reduction in PARP1 protein levels might be explained by the parallel reduction in Parp1 gene expression (Supplementary Fig. 5i). Given the role of PARP1 in DNA damage repair, we then evaluated DNA damage at the molecular level by quantifying cells with γH2AX and

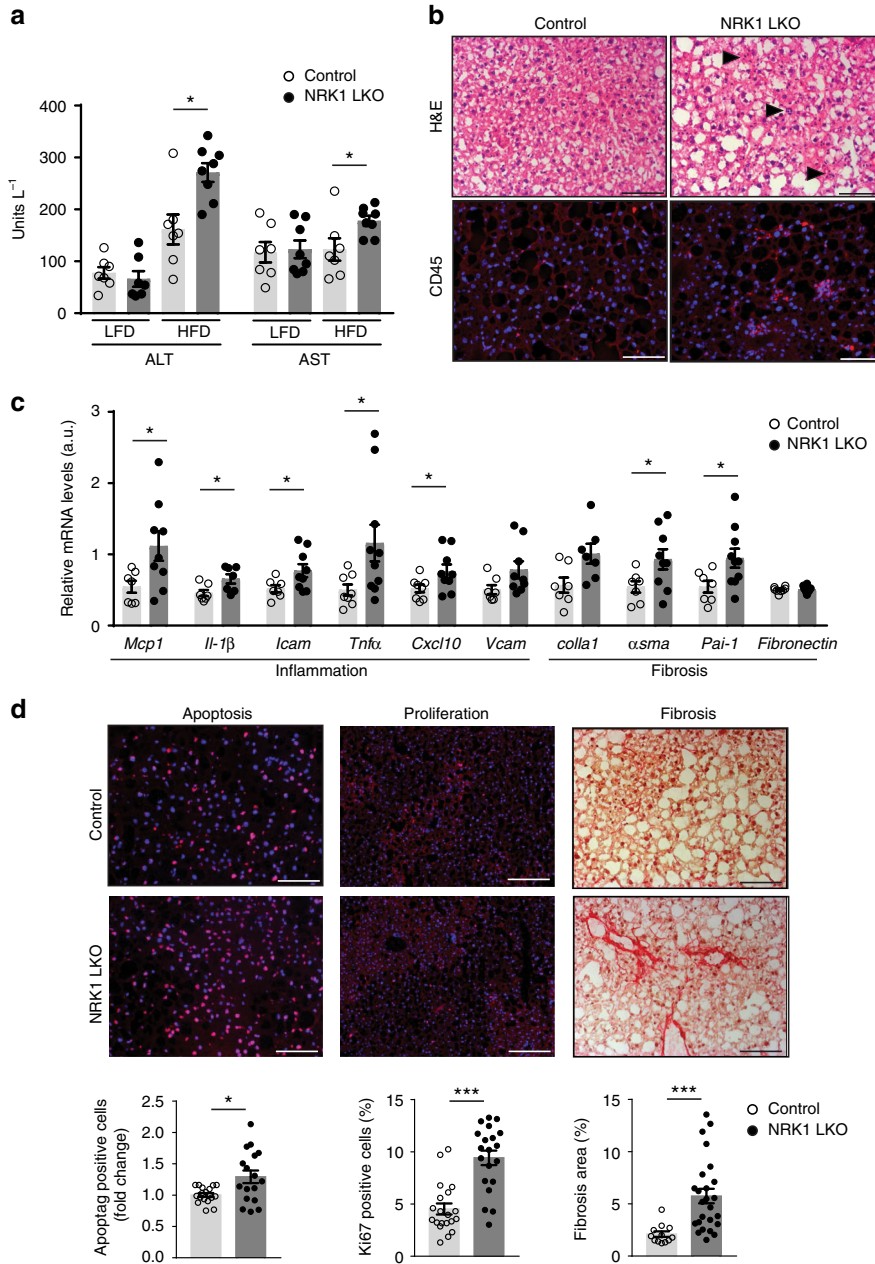

**Fig. 4** NRK1 liver-specific deletion fosters the development of NAFLD. **a** Plasma levels of liver damage markers, alanine transaminase (ALT) and aspartate transaminase (AST) in NRK1 LKO and control mice on LFD and HFD ($n = 7$ for all groups except NRK1 LKO on HFD, $n = 8$). **b** Representative H&E (top) and CD45 (bottom) staining on liver sections from NRK1 LKO and control mice on HFD showing immune cells infiltration (black arrows) (×20 magnification, scale bars: 100 μm). **c** Gene expression of markers of inflammation (left) and fibrosis (right) in the liver of NRK1 LKO mice and control mice on HFD ($n = 8$ for the control group; $n = 10$ for the NRK1 LKO group). **d** Immunohistochemistry on liver sections from NRK1 LKO and control mice on HFD. Representative staining (top) and quantification (bottom) for Apoptag (left), Ki-67 (center) (DAPI counterstaining) (×10 magnification, scale bars: 200 μm) and Sirius Red (right) (×20 magnification, scale bars: 100 μm) ($n = 4$ mice per group). Results shown are mean ± SEM, * and *** indicate $p < 0.05$ and $p < 0.001$, respectively, vs. the control group. The individual values and statistical tests used for each panel can be found in the Source Data file

53BP1 positive foci in the nucleus. The results illustrated a ~2.5-fold increase in the abundance of foci for both markers in the liver of NRK1 LKO mice compared to control littermates (Fig. 5e). Therefore, the reduced PARP1 activity in NRK1 LKO mice is not due to lower DNA damage. Rather, our results imply that the increased hepatic DNA damage induced by HFD triggers PARP1 activity in the liver[12,20], which generates a higher NAD⁺ demand that cannot be met in the NRK1 LKO mice. Therefore, this suggests that endogenous NR utilization is required to sustain

NAD⁺ levels and PARP1 activity in the liver of HFD-fed mice and that the failure to do so aggravates NAFLD.

Finally, we aimed to test whether restoring NAD⁺ levels in NRK1 LKO mice by supplementing with NRK1-independent NAD⁺ precursors could rescue their phenotype. To that end, NRK1 LKO mice were exposed to HFD and supplemented with or without NAM in drinking water for 12 weeks, since NAM-induced NAD⁺ synthesis is independent of NRK1 (Supplementary Fig. 6a). Surprisingly, hepatic NAD⁺ and NAM levels

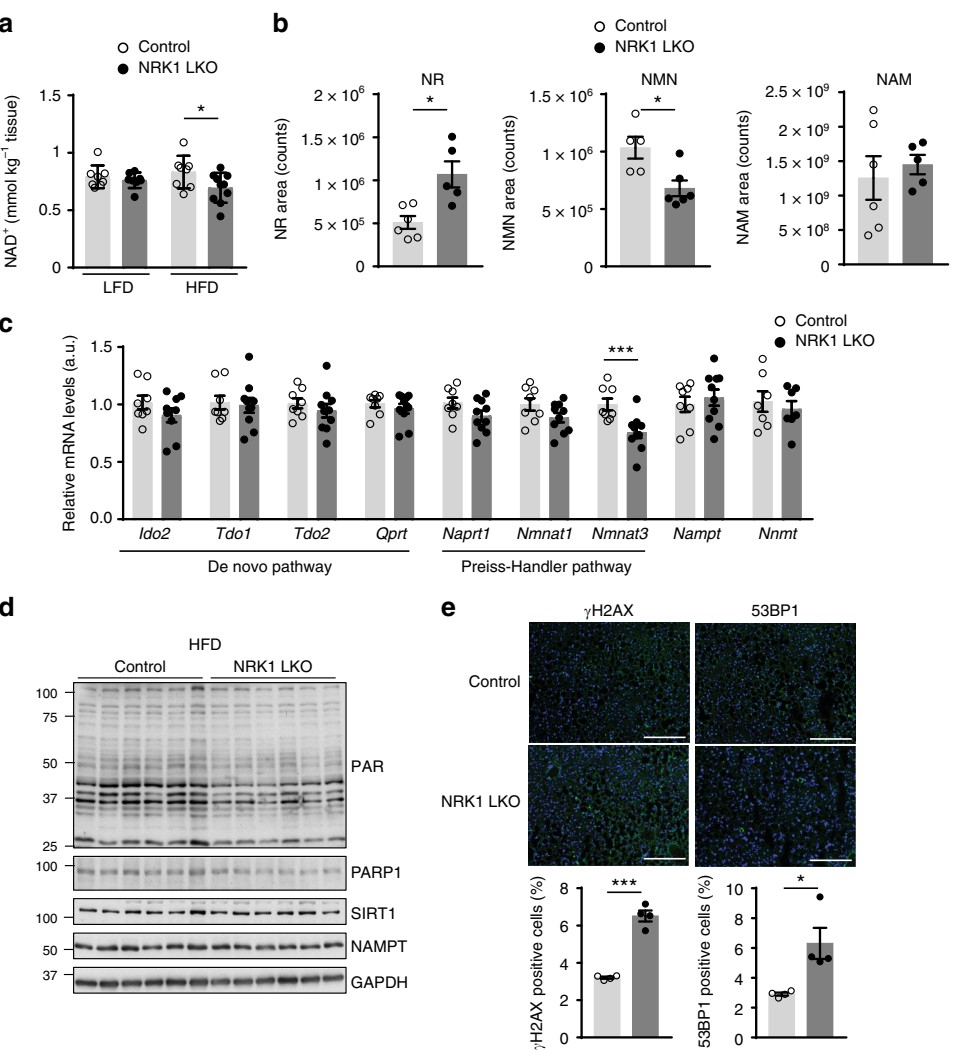

**Fig. 5** NRK1 deletion curbs PARP activity and exacerbates DNA damage. **a** NAD$^+$ levels in the liver of NRK1 LKO mice upon LFD and HFD ($n = 8$ for all groups except NRK1 LKO on HFD, $n = 10$; 24 weeks of age). **b** NR (left), NMN (center) and NAM (right) levels in the liver of NRK1 LKO and control mice upon HFD ($n = 6$ mice per group). **c** mRNA level of genes involved in the NAD$^+$ biosynthesis pathways ($n = 8$ mice for the control group, $n = 10$ mice for the NRK1 LKO group). **d** Levels of poly(ADP-ribose) protein modification, PARP1, SIRT1, NAMPT, and GAPDH in the liver of NRK1 LKO and control mice on HFD. **e** Representative staining (top) and quantification (bottom) of γH2AX (left) and 53BP1 (right) immunofluorescence on liver sections from NRK1 LKO and control mice on HFD ($n = 4$ mice per group, ×10 magnification, DAPI counterstaining, scale bars: 200 μm). Results shown are mean ± SEM, * and *** indicate statistical difference vs. control group at $p < 0.05$ and $p < 0.001$, respectively. The individual values and statistical tests used for each panel can be found in the Source Data file

remained similar between NAM and vehicle-treated groups (Supplementary Fig. 6b,c) indicating that NAM supplementation failed to increase absolute NAD$^+$ levels in the liver of NRK1 LKO mice. In contrast, we observed a 2.2-fold increase in 1-methylnicotinamide (MeNAM) and a 2.4-fold increase in MeXP levels in NAM-supplemented mice (Supplementary Fig. 6c). NAM supplementation in control mice did not increase NAM levels, but also increased MeNAM and MeXP, albeit to a lower degree than in NRK1 LKO mice. In particular, NAM supplementation led to a 1.8-fold increase in MeNAM (0.019 ± 0.002 for the Control-Veh group vs. 0.034 ± 0.002 for the Control-NAM group, expressed as mean ± SEM of peak area couns; $n = 7$ and $n = 6$, respectively) and a 1.8-fold increase in MeXP (0.023 ± 0.002 for the Control-Veh group vs. 0.043 ± 0.001 for the Control-NAM group, expressed as mean ± SEM of peak area counts; $n = 7$ ad $n = 6$, respectively).

NAM supplementation also prompted an increase in the mRNA level of *Nnmt* catalyzing the methylation of NAM in those animals (Supplementary Fig. 6d). At the phenotypical level,

NAM-supplemented NRK1-LKO mice displayed a slight decrease in body weight and a trend towards lower liver weight and hepatic TG content (Supplementary Fig. 6e–f). However, in line with the absence of effect on NAD$^+$ levels, NAM supplementation failed to have any significant impact on glucose tolerance, gluconeogenic capacity or mitochondrial respiration (Supplementary Fig. 6g–i). Nevertheless, we observed a modest decrease in γH2AX positive cells in NAM-supplemented NRK1 LKO mice compared to vehicle-treated mice (Supplementary Fig. 6j), suggesting a mild rescue of the PARP activity. Altogether, NAM supplementation did not significantly improve NAD$^+$ levels or mitochondrial dysfunction in NRK1 ablated livers, even if it conferred a modest protection against excessive DNA damage in NRK1 deficient mice.

## Discussion

NAD$^+$ can be derived from multiple precursors, however their individual contribution to the maintenance of intracellular

$NAD^+$ in mammalian tissues is poorly understood. NRK1, the enzyme initiating NR-mediated $NAD^+$ synthesis, is highly conserved in eukaryotes and abundantly expressed in hepatocytes. Thus, we hypothesized that $NAD^+$ synthesis from NR could have a physiological role in the liver. This work provides evidence that (1) endogenous NR metabolism is required to sustain hepatic $NAD^+$ levels in situations of metabolic damage and lipotoxicity and, (2) the inability to use NR as a $NAD^+$ precursor leads to mitochondrial dysfunction and amplifies the detrimental effects of HFD.

Hepatic NRK1 expression is not affected by HFD and NRK1 deficient mice does not exhibit glucose intolerance or insulin resistance on regular housing conditions. This clearly indicates that defective NR metabolism is not enough to trigger metabolic disease. Nevertheless, defects in gluconeogenesis are observed in NRK1 deficient mice, probably related to their marked hepatic mitochondrial dysfunction. The main cause for this mitochondrial respiratory deficit remains elusive, but cannot be attributed to lower mitochondrial content, as we did not observe changes in mitochondrial DNA content or respiratory complexes markers. Instead, it may be associated with impaired mitophagy or defective mitochondrial unfolded protein response, both of which are regulated by NR[8,21–25]. Although the observed phenotype is unlikely to originate from a major defect in mitophagy or mtUPR, investigating the exact contribution of NR metabolism to these processes would require an extensive study lying beyond the scope of this manuscript. The influence of NR on mitochondrial function has also been demonstrated in supplementation studies, where NR prevented the decline of mitochondrial function in liver, brown adipose tissue and skeletal muscle in situation of HFD and inherited mitochondrial diseases[10,26,27]. The compromised mitochondrial respiratory capacity in NRK1 LKO mice was associated with impaired FAO capacity. Notably, defects in FAO could also be observed in NRK1 deficient primary hepatocytes when they were challenged by low glucose medium or lipid overload. These observations, again, mirror the improved metabolic flexibility observed upon NR supplementation in rodents[10,28]. NRK1 LKO mice displayed a more marked glucose intolerance upon HFD. Paradoxically, NRK1 LKO mice were also characterized by lower fasting glycemia. This apparent contradiction can be explained by the different underlying causes for both phenomena. The reduced fasting glycemia likely stems from defective gluconeogenic capacity, probably related to mitochondrial dysfunction. The root of glucose intolerance, however, is insulin resistance and metabolic inflexibility, which compromises the shutdown of glucose production in response to a glucose/insulin challenge. There might be multiple reasons why NRK1 LKO mice develop insulin resistance. Exacerbated hepatic lipid accumulation and inflammation have all been linked to insulin resistance through different mechanisms, including the accumulation of detrimental lipid secondary species as well as the activation of inflammation-related signaling paths, such as JNK, IKK or novel PKCs[29]. Hence, insulin resistance in NRK1 LKO mice might not be necessarily consequent to a direct action of a $NAD^+$-dependent protein on insulin signaling, but secondary to the exacerbation of high-fat diet-induced steatosis. Nevertheless, it has been proposed that SIRT1 represses PTP1b transcription and allows sustaining insulin signaling[30], which could constitute a direct link between $NAD^+$ limitations and the impairment of insulin signaling. In agreement with our results in NRK1 deficient mice, depletion of hepatic $NAD^+$ pool via the expression of a dominant negative form of NAMPT also results in hepatic steatosis and insulin resistance[31]. Despite the specificity of NRK1 deletion in the liver, we cannot rule out a possible contribution of peripheral tissues to the whole-body insulin resistance observed in NRK1 LKO mice.

One of the main questions in the $NAD^+$ field is the existence of multiple pathways for $NAD^+$ synthesis and our lack of understanding on when, why and to what extent do they contribute to physiological $NAD^+$ homeostasis. As reported previously[13], NR metabolism seems to be dispensable to sustain $NAD^+$ levels in tissues from healthy, unchallenged young mice. In contrast, the loss of *Nampt* leads to a sharp depletion of $NAD^+$ levels in skeletal muscle and adipose tissues[32–34]. These observations advocate that $NAD^+$ salvage from NAM is the paramount mechanism maintaining basal $NAD^+$ levels in these tissues. Nevertheless, the effect of NAMPT inhibition on $NAD^+$ levels is less pronounced in hepatocytes than in myotubes[13], opening the possibility that NAMPT-independent pathways could significantly contribute to sustain $NAD^+$ levels in the liver. Accordingly, hepatic $NAD^+$ levels were diminished in NRK1 LKO mice upon HFD, when higher rates of hepatic DNA damage elicit PARP1 activation[12,19], prompting higher rates of $NAD^+$ consumption. Our work suggests that the inability to utilize NR limits $NAD^+$ availability for PARP1 activity, leading to DNA damage accumulation and a marked aggravation of NAFLD. So, why should NR be crucial for the maintenance of $NAD^+$ levels if $NAD^+$ synthesis could be channeled via the de novo pathway or from NAM? Our results do not support the hypothesis of an impaired expression of the enzymes belonging to NRK1-independent $NAD^+$ synthesis paths, even though we cannot fully rule out changes in activity due to post-translational modifications. Rather, the decline of NMN levels and accumulation of NR observed in NRK1 LKO livers, without alterations in NAMPT or NAM levels, suggests that there is no compensation through NAM to generate NMN and maintain hepatic $NAD^+$ in this situation. Accordingly, the administration of NAM to NRK1 LKO mice neither increased NMN and $NAD^+$ levels nor recovered their metabolic defects. Instead, the supplemented NAM seemed to be largely diverted towards clearance methylation/oxidation paths, as the hepatic levels of MeNAM and MeXP, as well as *Nnmt* expression, were increased after NAM supplementation (Supplementary Fig. 6b–d).

This exclusive need for NR could rely on the fact that the NR/NRK path is the only known $NAD^+$ biosynthetic pathway that allows $NAD^+$ synthesis without utilizing phosphoribosyl pyrophosphate (PRPP) as a ribosyl donor for $NAD^+$ synthesis, given that the ribose moiety is already part of the NR molecule[2]. Interestingly, hepatic PRPP content decreases upon HFD[35], which could limit the availability of PRPP for NAMPT, NAPRT or QPRT thereby compromising their enzymatic activities. This could be expected in situations characterized by a higher flux of PRPP through phosphoribosyltransferase activities of the purine/pyrimidine biosynthesis routes, such as in response to DNA damage or during hepatic regeneration. In these conditions, the ability of hepatocytes to generate $NAD^+$ from NRK1-independent pathways would be impaired thus NR would prevail as the major $NAD^+$ precursor. In this sense, NR supplementation has been shown to effectively improve liver regeneration[36]. As our mice were never supplemented with exogenous NR, our results also imply that either there is a dietary source of NR, that certain cell types might be able to produce and release NR - as it is the case in yeast[37,38] – or that NR could be produced by the gut microbiome. A final consideration is that we do not know the fate of accumulated NR in NRK1 LKO mice and its physiological consequences. Altogether, these results imply a physiological relevance of endogenous NR utlization in the maintenance of $NAD^+$ levels in mammalian cells and the potential pathophysiological conditions in which this might be relevant.

Strikingly, we did not observe major changes in sirtuin function in NRK1 LKO animals. Granted, our analyses only covered a

particular spectrum of sirtuins activities, and we cannot rule out time-, compartment- or target-specific changes in their activity. Nevertheless, SIRT1 overexpression in NAMPT deficient livers only partially corrected hepatic TG accumulation and ALT/AST levels[31] corroborating that defective SIRT1 activity is not fully accountable for the hepatic metabolic defects observed when $NAD^+$ biosynthesis is impaired. Instead, we detected severe impairment of PARP catalytic activity, as depicted by lower cellular PARylation levels. PARP1, the main PARP activity, is mainly regulated by DNA damage[2]. The decline of PARP activity in NRK1 LKO mice, however, cannot be explained by a reduced need for DNA repair, as they were burdened by enhanced liver DNA damage. Furthermore, we did observe reduced PARP1 protein levels as well as decreased Parp1 gene expression in NRK1 LKO mice suggesting that, in addition to an impaired enzymatic activity, a transcriptional feedback loop could also contribute to lower PARylation levels. Indeed, it has been reported that Parp1 expression could be controlled by a negative autoregulatory loop[39]. Hence, our data suggest that NRK1 deletion, by directly affecting $NAD^+$ pools and by triggering transcriptional regulation mechanisms, ultimately limits PARP1-dependant PARylation, resulting in defective DNA repair in response to hepatic lipotoxicity and thereby accumulating DNA damage. Of note, we detected a modest improvement on DNA damage and a trend towards lower steatosis upon NAM supplementation. This could be explained by the accumulation of MeNAM facilitated by higher Nnmt expression, which has been previously reported to have a direct effect in reducing liver TG[40], decreasing this way hepatic lipotoxicity.

Overall, this work provides evidence for the role of hepatic endogenous NR metabolism in the adaptability to nutritional challenges and protection against lipotoxicity. From an evolutionary perspective, NRK1 function could be critical in situation of high DNA damage or regeneration. Altogether, our results provide a context and a physiological relevance to NR utilization, and constitute a step further in our understanding of the complex interrelation among different $NAD^+$ precursors and biosynthesis paths.

## Methods

**Animal care and phenotyping**. All animal experiments were performed according to national Swiss and EU ethical guidelines and approved by the local animal experimentation committee under license VD 2770. NRK1 KO mice have been described previously[13]. NRK1 liver-specific KO (NRK1 LKO) mice were generated by crossing NRK1[loxP/loxP][13] with mice expressing the Cre recombinase under the albumin promoter, all in a pure C57BL/6NTac background (Taconic Biosciences). Mice were kept in a temperature- and humidity-controlled environment with a 12:12-h light-dark cycle. Mice had access to nesting materials and were provided with ad libitum access to water and commercial low- (LFD) or high-fat diet (HFD, %CHO/FAT/PROT:20/60/20), (D12450J and D12492, respectively, Research Diets Inc.). Metabolic phenotyping was performed on males after a 6-hr fasting unless otherwise stated and procedures can be found in the Supplementary Methods section.

**Histology**. H&E, Sirius Red and Oil Red O stainings were performed using the fully automated Ventana Discovery XT (Roche Diagnostics). Inflammation was assessed by immunofluorescence against CD45 marker. Fibrosis was assessed by quantifying Sirius Red stained area relative to controls in 5–6 microscopy fields at ×20 magnification per mouse ($n = 4$ per genotype) using a semi-automated image analysis of grayscale thresholded images using ImageJ software (NIH)[41]. Apoptosis was assessed by immunofluorescence using Apoptag Red in situ detection kit (Millipore S7165) and proliferation by immunofluorescence against Ki67 antibody on OCT-embedded sections. DNA damage in liver was assessed by immunofluorescence against γH2AX (Cell Signaling 9718, 1:100) and 53BP1 (NB100-304, Novus Biologicals, 1:500) on OCT-embedded sections. For IF quantification, 5–6 microscopy fields at ×10 (Ki67, γH2AX and 53BP1) or ×20 magnification (CD45, Apoptag) per mouse ($n = 4$ per genotype) and containing a minimum of 300 cells (DAPI counterstaining), were scored using ImageJ software.

**Respirometry studies**. Respirometry studies were performed in fresh liver homogenates or permeabilized soleus muscle using high-resolution respirometry

(Oroboros Oxygraph-2k, Oroboros Instruments), following the steps described in a recent publication[42].

**Molecular biology and Western blot**. Total mRNAs were extracted, processed and quantified as indicated in another manuscript[13]. The primers used are provided in the Supplementary Table 1. Gene expression was normalized with b2-microglobulin and cyclophillin. Protein extraction, quantification and western blotting procedures were performed as described previously[13]. All primary antibodies used are listed in Supplementary Table 2. Primary antibodies were used in 1:5000 dilution for GAPDH and 1:1000 dilution for others. Original uncropped blot images can be found in the online Supplementary Information file, together with quantifications for all western blot results using ImageJ software.

**Analysis of $NAD^+$ metabolites by LC–MS**. Sample preparation for metabolomics analysis was based on ref. [13]. Sample processing and analysis are detailed in the Supplemental Methods section.

**Statistical analyses**. Statistical analyses were performed with GraphPad Prism version 5.02 for Windows (La Jolla, CA, USA). In vitro analyses were performed as minimum in duplicates in three separate experiments. Differences between two groups were analyzed using a Student's two-tailed t-test. Two-way ANOVA analysis, classical (Fig. 1f) or for repeated measurements (all other cases) was used when comparing more groups, corrected with Tukey's post-hoc test (Fig. 1a). Given the explorative character of the analyses, all p-values are to be understood purely descriptively. Group variance, assessed using the F-test, was similar between groups. All data are expressed as mean ± S.E.M.

Additional experimental procedures can be found as Supplementary Information (Supplementary Methods section).

**Reporting summary**. Further information on research design is available in the Nature Research Reporting Summary linked to this article.

## Data availability

All data is available upon request. The source data underlying all bar and curves graphs can be found in the online Source Data File, as well as the specific statistical tests used in each graph. Original uncropped blots can also be found in the online Supplementary Information file, as well as quantifications for all western blot images.

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

## Acknowledgements

We thank members of the Cantó laboratory for helpful discussions, the members of the EPFL animal facility for technical support and the EPFL Histology Core Facility. M.V.-A. is supported by the EU Marie Skłodowska-Curie ITN-ChroMe (H2020-MSCA-ITN-Project number 675610).

## Author contributions

J.R., A.S., and C.C. designed the study. J.R., A.S., M.J., J.L.S-G., J.G.-G, M.V.-A, A.C., M.B., S.S.K. performed the experiments. A.V. provided statistical assistance for the analysis of results. M.P.G. and S.M. performed the analyses for NAD$^+$ metabolites. All authors analyzed data. J.R., A.S. and C.C. wrote the manuscript, which was approved by all authors.

## Additional information

**Competing interests:** All authors are employees of Nestlé Research S.A.

