## [Peer Review File · Nature Communications]

Editorial Note: This manuscript has been previously reviewed at another journal that is not operating a transparent peer review scheme. This document only contains reviewer comments and rebuttal letters for versions considered at Nature Communications. Parts of this Peer Review File have been redacted as indicated to maintain the confidentiality of unpublished data.

Reviewers' comments:

Reviewer #1 (Remarks to the Author):

In this revised manuscript, the authors have added data showing that exogenous NAM supplementation can not ameliorate defects in liver-specific NRK1 knockout mice, suggesting that the route by which hepatocyte NAD⁺ is important. The authors have also clarified multiple other points that strengthen their conclusions on the impact of endogenous NRK1 on liver metabolism. There are just a few remaining points that require clarification:

1) Why do HFD-fed NRK1 LKO mice show glucose intolerance? This is a fairly important point, as chow-fed NRK1 LKO mice have (perhaps paradoxically) slightly less glucose excursion in response to pyruvate. Does pyruvate tolerance in HFD-fed mice show the opposite phenotype? And if so, how does NRK1 regulate hepatic insulin sensitivity? Finally, although there does appear to be statistically significant differences in glucose during the GTT, calling these defects "severe" may be misleading for what appears to be a ~10% difference in AUC during the GTT.

2) In regards to other liver phenotypes, the authors convincingly show exacerbation of HFD-induced liver damage (by means of plasma AST/ALT and visualization of apoptotic cells) in NRK1 LKO mice. Characterization of these mice as having increased liver inflammatory cells based on H+E staining, however, is insufficient; this should be assessed by CD45 staining. Similarly, the Sirius Red staining provided is not convincing for fibrosis - what the image appears to show is vessel collagen staining, which is normal, as opposed to pericellular fibrosis that occurs in NASH. In addition, without accompanying methods, it's unclear how the authors quantitated these images. Finally, the image submitted for Reviewer use to represent hepatocyte ballooning is equally ambiguous. If the authors truly want to make these claims - which does not appear to be central to the point of this manuscript - these images should be adjudicated systematically, and if necessary, with assistance of a pathologist.

Reviewer #2 (Remarks to the Author):

The finding that NAM supplementation cannot suppress the NRK1KO phenotype is interesting, but there are a few changes that would be helpful the readers.

1. Administration of NAM to NRK1 LKO mice didn't increase NMN or NAD⁺ levels, nor did it recover the metabolic defects. Regarding the authors' statement, "One possibility to explain this exclusive need for NR is that the NRK path is the only biosynthesis pathway that can synthesize NAD⁺ without using phosphoribosyl pyrophosphate (PRPP)." The citation doesn't seem to explain this finding, nor do the authors attempt to give one. It would be helpful if the discussion includes a sentence or 2 where the phosphate might be coming from if not from PRPP.

2. The authors state, our "results also imply that either there is a dietary source of NR or that certain cell types might be able to produce and release NR, as it is the case in yeast." The authors may want to consider the possibility that the NR is generated in the gut by the microbiome.

3. The authors state, "Our results rule out the hypothesis of an impaired expression of the enzymes belonging to NRK1-independent NAD⁺ synthesis paths." The authors might want to consider that expression levels do not prove the pathway is active. Enzymes of course can be regulated at the post-translational level and it could be that NAMPT etc are simply rendered inactive even if they show up on a Western blot.

Reviewer #4 (Remarks to the Author):

The statistical analyses performed throughout this manuscript should be scrutinized. It is simply not correct to use one-way ANOVAs to test for differences when there are more than one factor. For instance, in cases where experiments included KO and WT mice on LFD and HFD, the data should be analyzed by a two-way ANOVA. Before correct statistical analyses have been applied throughout the manuscript, it is impossible for any reviewer to fully evaluate/appreciate the conclusions made by the authors.

It is not clear from the manuscript whether both male and female mice were used. Since this is a potential confounding effect, it is important to report this properly. Please indicate the gender of the mice in the methods section/legend of specific figures.

Please state which C57BL/6 substrain has been used for backcrossing.

NAM supplementation appears not to rescue the NRK1 KO phenotype. Given the finding of increased methylated NAM products, this could imply that NAMPT activity is limiting in the HFD setting. However, since NAMPT levels seem not to be affected by HFD feeding in the KO mice, and since higher levels of methylated NAM-related metabolites are observed, it is likely that NNMT pathway is induced. NNMT protein levels should be measured to further substantiate the claims made in the response to comments made by reviewer 2 (i.e., that "...under the high-fat feeding, NAM utilization is highly diverted towards methylation and oxidation, instead of NAD⁺ synthesis.>").

The authors present data on MeXP levels in both HFD-fed KO/WT mice and in KO/WT mice where the diet were supplemented with NAM. The relative increases in MeXP levels in the KO mice are similar in the two settings. What are the absolute levels of MeXP in the two experimental settings?

The authors state that the increased DNA damage induced by the HFD triggers PARP1 activity. This statement is backed up by a couple of references (i.e., 12 and 20). However, the WBs showing parylation are performed so only comparisons between genotypes within diets can be made. As the effect of NRK knockout on PARP activity in response to a HFD is completely central to 'story-line' presented, WBs should be done by including samples from all four conditions (i.e., WT-LFD, KO-LFD, WT-HFD, KO-HFD) next to each other on the same gel. Furthermore, whole-lane quantification should be performed.

The HFD induced hyperglycemia in the WT mice which was largely prevented in the KO mice. This effect occurred despite impaired glucose and insulin tolerance, hepatic steatosis, as well as reduced insulin signaling on the level of AKT in primary hepatocytes. The authors should discuss this seemingly contradictory finding. Furthermore, reporting AACs on relative values as done for the ITTs, when basal blood glucose levels are possibly different, will introduce a bias. For this reason, it is most appropriate to simply report the absolute levels - see the discussion on this in Ayala et al. "Standard operating procedures for describing and performing metabolic tests of glucose homeostasis in mice" *Dis Model Mech.* 2010 Sep-Oct; 3(9-10): 525-534. The determination of area over the curve will account for these differences in basal glucose levels since the baseline equals the basal glucose level.

Please provide all blood glucose values in the standard unit of mmol/L.

The authors show a slight decrease in NAD⁺ levels in KO mice fed a HFD. Given the significant reduction in mitochondrial function, it would be of interest to quantify the NAD⁺ pool in the mitochondria vs. nuclear/cytosolic. Isolation of mitochondria from liver can easily be done and data on

NAD⁺ in this compartment would add significantly to the understanding of the effects of Nrk1 knockout.

Letter to the reviewers:

Reviewer #1:

“In this revised manuscript, the authors have added data showing that exogenous NAM supplementation can not ameliorate defects in liver-specific NRK1 knockout mice, suggesting that the route by which hepatocyte NAD⁺ is important. The authors have also clarified multiple other points that strengthen their conclusions on the impact of endogenous NRK1 on liver metabolism. There are just a few remaining points that require clarification”

We thank the reviewer for his/her kind words and appreciation on the value of the new experiments.

“1) Why do HFD-fed NRK1 LKO mice show glucose intolerance? This is a fairly important point, as chow-fed NRK1 LKO mice have (perhaps paradoxically) slightly less glucose excursion in response to pyruvate.

This is indeed a critical point for discussion that has now been reinforced in the manuscript. The glucose intolerance of NRK1 LKO in high-fat diet fed mice stems from hepatic insulin resistance at the signaling level, while the impaired gluconeogenic capacity probably finds its root in mitochondrial dysfunction. Therefore, both aspects manifest uncoupled defects.

Hepatic insulin resistance will manifest as a failure to shut down glucose production in response to a glucose or insulin challenge. In the absence of a challenge, this will render the mice metabolically inflexible, but not necessarily hyperglycemic during fasting. Our view is that the hepatic glucose production tone in NRK1 LKO mice, albeit lower at the fasting baseline - will be unresponsive to insulin.

Does pyruvate tolerance in HFD-fed mice show the opposite phenotype?

We tried to perform PTTs in high-fat fed mice. However a 2 g/kg injection led to the almost immediate death in high-fat fed C57Bl/6NTac mice, irrespectively of the genotype. Mice only survived when we decreased the pyruvate load to 1 g/kg, yet at these low concentrations, we failed to see a major increase in glycemia or differences between genotypes.

And if so, how does NRK1 regulate hepatic insulin sensitivity?

NRK1 deficiency might impact on insulin sensitivity by direct and indirect means. From one side, deficient SIRT1 activity due to limited NAD⁺ could directly affect insulin signaling through PTP1b¹. This way, reduce SIRT1 activity would impede the full engagement of insulin effectors. Nevertheless, at the times examined we did not detect differences in SIRT1 activity. Therefore, it is very likely that the impact of NRK1 on insulin sensitivity is indirect and consequent to the higher hepatic lipid accumulation and deficient mitochondrial function, both of which have been largely related to insulin resistance. This is now also discussed in the manuscript.

Finally, although there does appear to be statistically significant differences in glucose during the GTT, calling these defects "severe" may be misleading for what appears to be a ~10% difference in AUC during the GTT."

We removed the word "severe", according to the reviewer's suggestion.

"2) In regards to other liver phenotypes, the authors convincingly show exacerbation of HFD-induced liver damage (by means of plasma AST/ALT and visualization of apoptotic cells) in NRK1 LKO mice. Characterization of these mice as having increased liver inflammatory cells based on H+E staining, however, is insufficient; this should be assessed by CD45 staining.

We now provide in addition to H&E staining, CD45 immunofluorescence data that show a 4-fold increase in CD45 stained area in NRK1 LKO mice compared to controls (Figure 4b). This result further supports our claims that liver inflammation is increased in NRK1 LKO mice originally based on H&E staining and mRNA analyses.

Similarly, the Sirius Red staining provided is not convincing for fibrosis - what the image appears to show is vessel collagen staining, which is normal, as opposed to pericellular fibrosis that occurs in NASH.

The referee is fully right on his/her observation and we now display more representative images for pericellular fibrosis (Fig. 4d), which was also prominent in NRK1 LKO mice on high-fat diet.

In addition, without accompanying methods, it's unclear how the authors quantitated these images.

We now provide additional methods on the quantification of images for fibrosis. Briefly, liver fibrosis was assessed by quantifying Sirius Red stained area in NRK1 LKO mice relative to controls in 5-6 microscopy fields at 20x magnification per mouse (n=4 per genotype) using a semi-automated image analysis of grayscale thresholded images using ImageJ software (NIH) where pixels of the detected regions were measured using the measure tool ².

Finally, the image submitted for Reviewer use to represent hepatocyte ballooning is equally ambiguous. If the authors truly want to make these claims - which does not appear to be central to the point of this manuscript - these images should be adjudicated systematically, and if necessary, with assistance of a pathologist."

As the referee notices, given the tangential nature of these observations, we decided to remove any mention of ballooning from the manuscript.

Reviewer #2:

"The finding that NAM supplementation cannot suppress the NRK1KO phenotype is

interesting, but there are a few changes that would be helpful the readers.

1. Administration of NAM to NRK1 LKO mice didn't increase NMN or NAD⁺ levels, nor did it recover the metabolic defects. Regarding the authors' statement, "One possibility to explain this exclusive need for NR is that the NRK path is the only biosynthesis pathway that can synthesize NAD⁺ without using phosphoribosyl pyrophosphate (PRPP)." The citation doesn't seem to explain this finding, nor do the authors attempt to give one. It would be helpful if the discussion includes a sentence or 2 where the phosphate might be coming from if not from PRPP."

We thank the reviewer for his/her comment. We have expanded this part to clarify the statement. PRPP provides the ribosyl group to the nicotinamide (or nicotinic acid) moiety of the NAD⁺ molecule. In the case of NR, ribose is already a part of the molecule, and the phosphate is provided by ATP through the NRK1-mediated phosphorylation.

"2. The authors state, our "results also imply that either there is a dietary source of NR or that certain cell types might be able to produce and release NR, as it is the case in yeast." The authors may want to consider the possibility that the NR is generated in the gut by the microbiome."

We thank the referee for this suggestion. This hypothesis is now mentioned in the discussion section.

"3. The authors state, "Our results rule out the hypothesis of an impaired expression of the enzymes belonging to NRK1-independent NAD⁺ synthesis paths." The authors might want to consider that expression levels do not prove the pathway is active. Enzymes of course can be regulated at the post-translational level and it could be that NAMPT etc are simply rendered inactive even if they show up on a Western blot."

According to the reviewer's comment, we now added in the discussion section a sentence stating that we cannot rule out changes in enzymatic activity due to post-translational modifications.

Reviewer #4:

"The statistical analyses performed throughout this manuscript should be scrutinized. It is simply not correct to use one-way ANOVAs to test for differences when there are more than one factor. For instance, in cases where experiments included KO and WT mice on LFD and HFD, the data should be analyzed by a two-way ANOVA. Before correct statistical analyses have been applied throughout the manuscript, it is impossible for any reviewer to fully evaluate/appreciate the conclusions made by the authors."

We thank the reviewer's comment, as this made us notice that our statistics section was not properly stated. Following the referee's comment we have now carefully revised the statistical tests used for each figure and this is now better specified in the methods section.

“It is not clear from the manuscript whether both male and female mice were used. Since this is a potential confounding effect, it is important to report this properly. Please indicate the gender of the mice in the methods section/legend of specific figures.”

We exclusively used male mice, as C57Bl/6 female mice rarely develop obesity and insulin resistance upon high-fat feeding. This is now specified in the methods section.

“Please state which C57Bl/6 substrain has been used for backcrossing.”

We now specify in the methods section how the mice have been backcrossed and used exclusively in C57Bl/6NTac background.

“NAM supplementation appears not to rescue the NRK1 KO phenotype. Given the finding of increased methylated NAM products, this could imply that NAMPT activity is limiting in the HFD setting. However, since NAMPT levels seem not to be affected by HFD feeding in the KO mice, and since higher levels of methylated NAM-related metabolites are observed, it is likely that NNMT pathway is induced. NNMT protein levels should be measured to further substantiate the claims made in the response to comments made by reviewer 2 (i.e., that “..under the high-fat feeding, NAM utilization is highly diverted towards methylation and oxidation, instead of NAD⁺ synthesis.”).”

This is an excellent comment from the referee. We have now analyzed *Nnmt* expression level in the liver of NRK1 LKO mice on high fat diet and supplemented with NAM or its vehicle. Our results indicated that *Nnmt* expression in NRK1 LKO mice is increased ~1.5 fold upon NAM supplementation (Supplementary Fig. 6d). However, we did not observe any change in NRK1 LKO mice compared to their control littermates on HFD and in the absence of any supplementation (Fig. 5c). Notably, we also detected an increased *Nnmt* expression in the liver of control mice under HFD that could contribute to higher MeXP levels (Supplementary Fig. 5c). Altogether, this indicates that NAM accumulation in the supplemented NRK1 LKO mice could drive the expression of *Nnmt*. The increase in *Nnmt* could further facilitate the shuttling of NAM towards excretion routes, as now discussed in the manuscript.

“The authors present data on MeXP levels in both HFD-fed KO/WT mice and in KO/WT mice where the diet were supplemented with NAM. The relative increases in MeXP levels in the KO mice are similar in the two settings. What are the absolute levels of MeXP in the two experimental settings?”

Unfortunately, our NAD⁺-metabolite measurement methods are only semi-quantitative, allowing us to compare between groups run in a parallel experiment, but not between separately measured experiments. Therefore, and unfortunately, we cannot provide absolute levels of MeXP.

“The authors state that the increased DNA damage induced by the HFD triggers PARP1 activity. This statement is backed up by a couple of references (i.e., 12 and 20). However, the WBs showing parylation are performed so only comparisons between genotypes within diets can be made. As the effect of NRK knockout on PARP activity in response to a HFD is completely central to ‘story-line’ presented, WBs should be done

by including samples from all four conditions (i.e., WT-LFD, KO-LFD, WT-HFD, KO-HFD) next to each other on the same gel. Furthermore, whole-lane quantification should be performed.”

The influence of high-fat diet on liver PARylation has been extensively studied in previous studies. All of them consistently reported that hepatic PARP activity is increased by high-fat diets³⁻⁵, as well as by alcohol-induced fatty liver⁶. For this reason we mostly focused our PARP activity analyses on the genotype comparisons. Nevertheless, we have now clarified that this evidence comes from previous studies.

As requested by the reviewer, we now provide whole lane quantifications by profiling the different band peaks along the lane (Supplementary information: Western Blot quantification Fig. 5d and Suppl. Fig. 5g)

“The HFD induced hyperglycemia in the WT mice which was largely prevented in the KO mice. This effect occurred despite impaired glucose and insulin tolerance, hepatic steatosis, as well as reduced insulin signaling on the level of AKT in primary hepatocytes. The authors should discuss this seemingly contradictory finding.”

We agree with the reviewer’s comment and we discuss this point more extensively in the discussion section of the revised manuscript, as specified before in response to Reviewer 1.

“Furthermore, reporting AACs on relative values as done for the ITTs, when basal blood glucose levels are possibly different, will introduce a bias. For this reason, it is most appropriate to simply report the absolute levels - see the discussion on this in Ayala et al. “Standard operating procedures for describing and performing metabolic tests of glucose homeostasis in mice” Dis Model Mech. 2010 Sep-Oct; 3(9-10): 525-534. The determination of area over the curve will account for these differences in basal glucose levels since the baseline equals the basal glucose level.”

Curves with absolute values for ITT tests are now provided and substituted to relative values graph in Fig. 2g, Supplementary Fig. 1b and 2d.

“Please provide all blood glucose values in the standard unit of mmol/L.”

All blood glucose values are now presented in mmol/L.

“The authors show a slight decrease in NAD⁺ levels in KO mice fed a HFD. Given the significant reduction in mitochondrial function, it would be of interest to quantify the NAD⁺ pool in the mitochondria vs. nuclear/cytosolic. Isolation of mitochondria from liver can easily be done and data on NAD⁺ in this compartment would add significantly to the understanding of the effects of Nr1h3 knockout.”

This is an interesting suggestion from the referee. We have now measured mitochondrial NAD⁺ in our samples. Mitochondrial NAD⁺ content was similar between genotypes (Supplementary Fig. 5a). This might explain why we observed no differences in SIRT3 activity and global mitochondrial acetylation levels. In turn, this also implies that the decrease in NAD⁺ must be fully attributed to losses in the nuclear/cytosolic compartment. These results go in line with the concept proposed by Yang et al.⁷, in which NAD⁺ levels are independently regulated in different cellular

compartments and that, even upon forced NAD⁺ depletion, cells will strive to maintain mitochondrial NAD⁺ levels in order to survive.

REFERENCES

- 1 Sun, C. *et al.* SIRT1 improves insulin sensitivity under insulin-resistant conditions by repressing PTP1B. *Cell Metab* **6**, 307-319 (2007).
- 2 Schipke, J. *et al.* Assessment of cardiac fibrosis: a morphometric method comparison for collagen quantification. *Journal of applied physiology (Bethesda, Md. : 1985)* **122**, 1019-1030 (2017).
- 3 Gariani, K. *et al.* Inhibiting poly ADP-ribosylation increases fatty acid oxidation and protects against fatty liver disease. *Journal of hepatology* **66**, 132-141 (2017).
- 4 Huang, K. *et al.* PARP1-mediated PPARalpha poly(ADP-ribosyl)ation suppresses fatty acid oxidation in non-alcoholic fatty liver disease. *Journal of hepatology* **66**, 962-977 (2017).
- 5 Pang, J. *et al.* Inhibition of Poly(ADP-Ribose) Polymerase Increased Lipid Accumulation Through SREBP1 Modulation. *Cell Physiol Biochem* **49**, 645-652 (2018).
- 6 Zhang, Y. *et al.* Inhibition of Poly(ADP-Ribose) Polymerase-1 Protects Chronic Alcoholic Liver Injury. *Am J Pathol* **186**, 3117-3130 (2016).
- 7 Yang, H. *et al.* Nutrient-sensitive mitochondrial NAD⁺ levels dictate cell survival. *Cell* **130**, 1095-1107 (2007).

Reviewers' comments:

Reviewer #1 (Remarks to the Author):

The authors have addressed all my comments and concerns.

Reviewer #2 (Remarks to the Author):

-

Reviewer #4 (Remarks to the Author):

The authors have significantly improved the manuscript by clarifying some of my previous points and by providing additional data to support their conclusions. Given the confusion regarding the statistical analyses performed for each data set, it was not possible to properly evaluate the data presented in the previous version of the manuscript. Thus, there are now several issues that remain to be explained/corrected.

The conclusions from the first part of the results section describing data from whole body NRK1 knockout mice appear weak and do not clearly contribute to the remaining part of the manuscript. The link between the reduced gluconeogenic capacity and mitochondrial function is speculative and also poorly discussed in the Discussion. The model is sub-optimal given the global knockout strategy, and some experiments have been performed with only a very limited number of mice. For example, the measurement of glycemia and mitochondrial function the n is only 4. It is doubtful that prior power calculations suggested to use this low number of animals. Moreover, data on protein abundance and mRNA expression are shown for 6-7 animals. Are different cohorts of mice studied, or are all data from the same cohort? If so, why were not all animals used? LFD-fed NRK1 LKO mice show minor reductions in respiratory capacity which were intensified in the HFD-fed NRK1 LKO mice. However, since NRK1 KO mice were not studied on a HFD, the defect in respiratory capacity may be related to other confounding factors that were not controlled for. Moreover, it is not clear from the manuscript whether the O₂ flux data shown in Supplementary Figure 4A-B were performed in fasted or fed mice? Finally, given the fact that over-night fasting is a severe metabolic challenge for mice and is likely to induce torpor and resulting changes in metabolism (see for instance PMID 30040481 and 24025567), it is possible that the data obtained from the NRK1 KO mice are confounded by the long fast. My recommendation will therefore be to remove all data from this strain from the manuscript.

As discussed in the manuscript (lines 310-314), the impaired mitochondrial function may be related to changes in mitophagy or mitochondrial unfolded protein response. However, given that the papers referenced did not use genetic models to affect expression of Nmrk1, but applied NR in the different contexts, it would strengthen the current manuscript if reason(s) for the impaired mitochondrial function were explored further. It would be relatively easy to assess protein/mRNA markers for mitophagy and UPRmt. It is important to provide a possible explanation given the new data of unchanged mitochondrial NAD⁺ levels now included in the manuscript (Supplementary Figure 5A).

In the Supplementary Methods it is stated that specific bands from Western blots were normalized to bands obtained for GAPDH. It is a misconception that normalization to a "house-keeping" protein will provide a more accurate quantification. In fact, this approach will potentially be a source of increased variation because quantification of the house-keeping protein is also associated with a certain amount

of variability. Moreover, it is exceedingly difficult to choose a single suitable house-keeping protein as abundance of this protein may be affected differently between the different genetic models. Thus, results from all the included Western blots should be presented without normalization. Scrutinizing the Western blots, I noticed that the authors quantify a band for SIRT1 running above 100 kDa (Figure 5D). However, in Supplementary Figure 2A (both panels), the most prominent band for SIRT1 runs below 100 kDa. For all blots containing multiple bands, the authors should indicate the specific band with an arrow.

As mentioned in the first round of comments to the authors, and irrespective of what others have found previously, it is important to show how parylation is affected with diet and genotype in the present context. Thus, a Western blot with samples from all conditions loaded next to each other should be presented. This blot can be presented in the supplementary material.

In Supplementary Figure 6C the authors present data on NAM, MeNAM, and MeXP from NRK1 LKO mice. Although it is not really clear from the section on "Animal phenotyping", it must be assumed that both WT and LKO mice were treated with NAM. Thus, data from both genotypes should be presented.

Letter for the referees:

We thank all the referees for their constructive criticism during the revision process. We now provide a revised version of our manuscript addressing the few remaining issues raised by one of the reviewers.

Referee 4:

“The conclusions from the first part of the results section describing data from whole body NRK1 knockout mice appear weak and do not clearly contribute to the remaining part of the manuscript. The link between the reduced gluconeogenic capacity and mitochondrial function is speculative and also poorly discussed in the Discussion. The model is sub-optimal given the global knockout strategy, and some experiments have been performed with only a very limited number of mice. For example, the measurement of glycemia and mitochondrial function the n is only 4. It is doubtful that prior power calculations suggested to use this low number of animals. Moreover, data on protein abundance and mRNA expression are shown for 6-7 animals. Are different cohorts of mice studied, or are all data from the same cohort? If so, why were not all animals used?”

As the referee guesses, these were different cohorts. In this case, the respirometry analyses the referee refers to correspond to a question raised in a previous revision round requesting a feeding/fasting comparison on respirometry analyses. Respirometry analyses cannot be performed on frozen tissues, so we needed new mice to perform the experiment. Hence this is why the n is different from other analyses and why previous animals/frozen samples could not be used for this purpose. This, however, does not compromise the robustness of the results.

“LFD-fed NRK1 LKO mice show minor reductions in respiratory capacity which were intensified in the HFD-fed NRK1 LKO mice. However, since NRK1 KO mice were not studied on a HFD, the defect in respiratory capacity may be related to other confounding factors that were not controlled for.”

We now show respirometry data for the HFD on the whole body knockout mice too. As can be seen in the new Supplemental Figure 4C, NRK1 null mice also suffer from deficient mitochondrial respiratory capacity. The effects are slightly more severe on the total KO mice, which suggests that the phenotype is partially, but not totally, caused by the liver NRK1 deletion.

“Moreover, it is not clear from the manuscript whether the O₂ flux data shown in Supplementary Figure 4A-B were performed in fasted or fed mice?”

These experiments were performed in mice fasted for 6 hrs. This is fasted at 8 am and sacrificed at 14 hrs. This is now stated in the figure legend.

“Finally, given the fact that over-night fasting is a severe metabolic challenge for mice and is likely to induce torpor and resulting changes in metabolism (see for instance PMID 30040481 and 24025567), it is possible that the data obtained from the NRK1 KO mice are confounded by the long fast. My recommendation will therefore be to remove all data from this strain from the manuscript.”

Most of the experiments in the manuscript were done after a 6-hr fast, except where otherwise indicated (e.g.: ipGTT experiments). This is now clarified in the methods

section (Page 20). A 6-hrs fast is not a severe intervention for mice and, hence, we do not feel these data should be removed from the manuscript.

“As discussed in the manuscript (lines 310-314), the impaired mitochondrial function may be related to changes in mitophagy or mitochondrial unfolded protein response. However, given that the papers referenced did not use genetic models to affect expression of Nmrk1, but applied NR in the different contexts, it would strengthen the current manuscript if reason(s) for the impaired mitochondrial function were explored further. It would be relatively easy to assess protein/mRNA markers for mitophagy and UPRmt. It is important to provide a possible explanation given the new data of unchanged mitochondrial NAD⁺ levels now included in the manuscript (Supplementary Figure 5A).”

We respectfully disagree with the referee on the “relatively easy” nature of these experiments. The proper evaluation of mitophagy fluxes and mtUPR responses are demanding experimental expeditions, especially *in vivo*. They require, among other set ups, the elucidation of a positive control situation, the study of a dynamic range in terms of timings and amplitude in response to stimuli, and, in the case of mitophagy, dedicated imaging techniques (from EM to fluorescence). Such a study in our mouse models could take years and would constitute the subject of a manuscript on its own.

For the interest of the referee, we analyzed baseline levels of mitophagy or mtUPR markers, with no major changes in any of the cases, albeit a modest decrease in mitochondrial Parkin levels was observed in our HFD-fed LKO mice. However, these data should be taken cautiously, as western blots are only semiquantitative and baseline levels do not testify for dynamic responses.

[Redacted]

a. Protein level of Hsp60, Clpp and GAPDH in whole protein lysates from liver of NRK1 LKO and control mice on HFD (n=6 mice per group). Quantification by normalization to GAPDH. **b.** Protein level of Parkin, SQSTM1/p62 and Porin in mitochondrial fractions from liver of NRK1 LKO and control mice on HFD (n=6 mice per group). Quantification by normalization to Porin.

Given the relatively minor phenotype of our mice on LFD, it is unlikely that NRK deficiency leads to major, structural, changes in baseline mtUPR or mitophagy-related proteins. Instead, the phenotype rather stems for cumulative changes based on small differences in the amplitude of the response to particular stimuli (feeding, stress, etc.). We have slightly modified this point of the discussion to highlight that 1/ a proper study on mitophagy/mtUPR lies beyond the scope of this manuscript and 2/ that the phenotype is unlikely to derive from a structural/major defect in mitophagy or mtUPR (Page 15).

“In the Supplementary Methods it is stated that specific bands from Western blots were normalized to bands obtained for GAPDH. It is a misconception that normalization to a “house-keeping” protein will provide a more accurate quantification. In fact, this approach will potentially be a source of increased variation because quantification of the house-keeping protein is also associated with a certain amount of variability. Moreover, it is exceedingly difficult to choose a single suitable house-keeping protein as abundance of this protein may be affected differently between the different genetic models. Thus, results from all the included Western blots should be presented without normalization.”

Following the advice of the Nature Communications editorial board, we will maintain normalization on our western blot analyses.

“Scrutinizing the Western blots, I noticed that the authors quantify a band for SIRT1 running above 100 kDa (Figure 5D). However, in Supplementary Figure 2A (both panels), the most prominent band for SIRT1 runs below 100 kDa. For all blots containing multiple bands, the authors should indicate the specific band with an arrow.”

We thank the referee for spotting this potential source of confusion. The correct band for SIRT1 is the one slightly above 100 kDa. This evidence comes from works with SIRT1 transgenic mice, where only the top band is affected by the transgenesis¹. To clarify this point for the reader, and as suggested by the referee, we added an arrow next to the correct band.

“As mentioned in the first round of comments to the authors, and irrespective of what others have found previously, it is important to show how parylation is affected with diet and genotype in the present context. Thus, a Western blot with samples from all conditions loaded next to each other should be presented. This blot can be presented in the supplementary material.”

Unfortunately, we have to respectfully disagree again with the referee. Our sample collection conditions (e.g.: timing) was optimized to interrogate genotype differences, but not diet-related differences. These two questions require separate approaches, as high-fat feeding alters the feeding behavior of mice and feeding itself influences PARP activity in the liver². Further, our sampling timings do not match other publications where the influence of diet on PARylation was analyzed, making the comparison with previous literature impossible. Finally, such analyses would force us to re-utilize samples, knowing that, as with any other post-translational modification, freezing/thawing cycles compromise PARylation.

The fact that neither the samples nor the conditions for such analysis will be appropriate to obtain a solid conclusion should immediately disqualify the option of performing the test. Irrespectively of the outcome, the result would be flawed in

nature and not apt for publication (in neither former nor supplemental figure). We also feel that generating a new mouse cohort for this tangential question is definitely not justified from scientific/ethical perspectives. As stated multiple times, the goal of this manuscript is to evaluate the physiological impact of NR utilization on metabolic and molecular parameters, not the impact of HFD on PARP activity. For the later point, we refer the referee and the readers to previous works demonstrating increased PARP activity in tissues from HFD-fed rodents and in obese humans³⁻⁸.

“In Supplementary Figure 6C the authors present data on NAM, MeNAM, and MeXP from NRK1 LKO mice. Although it is not really clear from the section on “Animal phenotyping”, it must be assumed that both WT and LKO mice were treated with NAM. Thus, data from both genotypes should be presented.”

As requested by the referee. We now mention in the text the levels of NAM, meNAM and MeXP after NAM supplementation (Page 13).

REFERENCES

- 1 Boutant, M. *et al.* SIRT1 enhances glucose tolerance by potentiating brown adipose tissue function. *Molecular metabolism* **4**, 118-131 (2015).
- 2 Asher, G. *et al.* Poly(ADP-ribose) polymerase 1 participates in the phase entrainment of circadian clocks to feeding. *Cell* **142**, 943-953 (2010).
- 3 Rappou, E. *et al.* Weight Loss Is Associated With Increased NAD(+)/SIRT1 Expression But Reduced PARP Activity in White Adipose Tissue. *The Journal of clinical endocrinology and metabolism* **101**, 1263-1273 (2016).
- 4 Gariani, K. *et al.* Inhibiting poly ADP-ribosylation increases fatty acid oxidation and protects against fatty liver disease. *Journal of hepatology* **66**, 132-141 (2017).
- 5 Mukhopadhyay, P. *et al.* PARP inhibition protects against alcoholic and non-alcoholic steatohepatitis. *Journal of hepatology* **66**, 589-600 (2017).
- 6 Bai, P. *et al.* PARP-1 inhibition increases mitochondrial metabolism through SIRT1 activation. *Cell Metab* **13**, 461-468 (2011).
- 7 Mohamed, J. S., Hajira, A., Pardo, P. S. & Boriek, A. M. MicroRNA-149 inhibits PARP-2 and promotes mitochondrial biogenesis via SIRT-1/PGC-1alpha network in skeletal muscle. *Diabetes* **63**, 1546-1559 (2014).
- 8 Huang, K. *et al.* PARP1-mediated PPARalpha poly(ADP-ribosylation) suppresses fatty acid oxidation in non-alcoholic fatty liver disease. *Journal of hepatology* **66**, 962-977 (2017).

REVIEWERS' COMMENTS:

Reviewer #4 (Remarks to the Author):

I went through some of the data in the source file. Based on my analyses of the data, I would recommend the authors to seek support from a statistician. I believe that some of the conclusions made by the authors should be adjusted and the text updated. In the following I go through four examples where I believe the authors have reached incorrect conclusions.

Figure 1A: The data in this figure should be analyzed by a repeated two-way ANOVA as I have done below (in SPSS). It is clear from the output (i.e., Tests of Within-Subjects Effects) that while there is a clear main effect of time, there is no time x genotype interaction present. Moreover, from the other output (i.e., Test of Between-Subjects Effects) there are no genotype effect. Thus, the star that is presently indicating a difference between genotypes is not valid.

Tests of Within-Subjects Effects

Source	Type III Sum of Squares	df	Mean Square	F	Sig.
Time	459,494	6	76,582	134,135	,000
Time * Genotype	5,697	6	949	1,663	,144
Error(Time)	37,682	66			,571

Tests of Between-Subjects Effects

Source	Type III Sum of Squares	df	Mean Square	F	Sig.
Intercept	6012,243	1	6012,243	1907,115	,000
Genotype	10,455	1	10,455	3,316	,096
Error	34,678	11			3,153

Figure 1B: This data set is blood glucose measurements from the same animals taken at three different time points. Thus, the statistical test to use is again a repeated two-way ANOVA. The authors indicate in the legend that 4 animals were used for this experiment. However, data for n=12-13 are shown in the source data file. Moreover, when the data are analyzed (in SPSS) there are no time x genotype interaction in the data set and no genotype effect. Thus, it is not correct to add the two stars at T6 as done by the authors in the present version of the manuscript.

Tests of Within-Subjects Effects

Source	Type III Sum of Squares	df	Mean Square	F	Sig.
Time	191,140	2	95,570	193,497	,000
Time * Genotype2	3,141	2	1,571	3,180	,051
Error(Time)	22,720	46			,494

Tests of Between-Subjects Effects

Source	Type III Sum of Squares	df	Mean Square	F	Sig.
Intercept	4129,327	1	4129,327	4568,986	,000
Genotype2	1,713	1	1,713	1,896	,182
Error	20,787	23			,904

Figure 1F: Here I only tested the CI+CII data set from fed and fasted WT and KO animals. The test to use is a normal two-way ANOVA. As indicated below (the output is from SigmaPlot) there is a main effect of genotype and no other statistical significant effects. Thus, the authors cannot differentiate

between the fed and fasted condition as they have done now. The correct way to include the statistical indicators would be to add three stars above both black bars. Finally, the authors indicate in the legend that n=4 animals were used for these experiment. However, they include data from 5-6 animals in the source data file.

Normality Test (Shapiro-Wilk): Passed (P = 1,000)

Equal Variance Test (Brown-Forsythe): Passed (P = 0,612)

Source of Variation	DF	SS	MS	F	P
Genotype	1	1529,385	1529,385	19,335	<0,001
Fed-fasted	1	114,995	114,995	1,454	0,243
Genotype x Fed-fasted	1	52,933	52,933	0,669	0,423
Residual	19	1502,873	79,099		
Total	22	3264,710	148,396		

Figure 5A: Again, a straight-forward two-way ANOVA should be applied. The output (from SigmaPlot) shows a main effect of genotype. Thus, one star above the black bars should be included in the figure, and the statistical indicator in the present version of the figure is not correctly placed. The interpretation of the data (lines 228-229) is clearly not supported by the statistics and the text should be revised.

Normality Test (Shapiro-Wilk): Passed (P = 0,820)

Equal Variance Test (Brown-Forsythe): Passed (P = 0,356)

Source of Variation	DF	SS	MS	F	P
Genotype	1	0,0589	0,0589	4,498	0,042
Diet	1	0,00121	0,00121	0,0926	0,763
Genotype x Diet	1	0,0247	0,0247	1,884	0,180
Residual	30	0,393	0,0131		
Total	33	0,484	0,0147		

Reviewer #5 (Remarks to the Author):

Endogenous Nicotinamide Riboside metabolism protects against diet-induced liver damage seems to be a well-written article.

But, I am a statistician and this was the reason I was consulted. Since I have little knowledge of the actual subject of the article, I will only refer to the statistical aspects of the article in my review. The "Statistical analyses" section of the article is rather brief, but this is also due to the fact that the statistical methods used in the analysis of this paper are purely standard procedures, appropriate to the small number of cases in the samples. The sample sizes are mostly 6-7, a maximum of 10 mice was used. This is standard in such experiments. However, this implies that such an analysis is purely explorative and has no confirmatory character, which means that hypothesis tests and especially p-

values should not be overinterpreted. Due to the small number of cases, the power, i.e. the probability of rejecting a hypothesis if an effect is given, is very low. Hypothesis tests are always constructed in such a way that one should never infer the confirmation of the hypothesis from a non-significance, but in such a case this is particularly critical. In the present article, at one point or another, this is happening. For example on p.6, l.116 "... did not influence skeletal muscle respiratory ..." or on p. 7, l. 146 "... was not due to alteration of the gluconeogenic gene expression...".

One could discuss whether a non-parametric test, i.e. a Mann-Whitney-U-test, should be used instead of the t-tests due to the small number of cases. The article does not mention anything about the normal distribution assumption of the examined variables. However, due to the fact that I would only consider such an analysis as presented here to be explorative anyway, this is not strictly necessary. Additionally, the t-test is still quite robust even against violations against the normal distribution assumption.

In general, the article should focus more on effect sizes and standard deviations (also due to the explorative character already explained above), these are completely missing in the actual article and can only be read approximatively from the graphics.

Overall, a large number of hypothesis tests were performed. The authors were apparently aware that this could lead to a multiple test problem, but they only adjusted for multiple testing within individual settings. In view of the very large number of tests performed overall, one is nevertheless confronted here with a high type I error inflation. In my opinion, multiple testing can be dispensed with completely, in return it must be made clear that all p-values are to be understood purely descriptively. This also reflects the explorative character of the analysis. Again, this should be made clearer in the course of the article.

Rebuttal letter:

Referee 4:

We thank the referee for the comments on our manuscript, which helped us identifying some omissions. This said, there are several points from the referee with which we respectfully disagree. A point per point answer is provided below.

“I went through some of the data in the source file. Based on my analyses of the data, I would recommend the authors to seek support from a statistician. I believe that some of the conclusions made by the authors should be adjusted and the text updated. In the following I go through four examples where I believe the authors have reached incorrect conclusions.”

As suggested by the reviewer, we did consult with a biostatistician for the statistical analysis of our data. We performed additional analysis and tests and compiled all the data, statistical tests used and results, in the updated data source file. We updated the figures and amended our text accordingly. Moreover, we provide a point-by-point answer to the specific points raised by the reviewer below. Most notably, none of the analyses resulted in substantial changes in our conclusions.

“Figure 1A: The data in this figure should be analyzed by a repeated two-way ANOVA as I have done below (in SPSS). It is clear from the output (i.e., Tests of Within-Subjects Effects) that while there is a clear main effect of time, there is no time x genotype interaction present. Moreover, from the other output (i.e., Test of Between-Subjects Effects) there are no genotype effect. Thus, the star that is presently indicating a difference between genotypes is not valid.”

When assessing the difference between genotypes in the course of a Pyruvate Tolerance Test, our conclusion is based on the difference between Areas Under the Curve (AUCs), calculated with a t-test. We agree with the reviewer that a 2-way ANOVA analysis (for repeated measurements) on the glycemia values did not show any effect of the genotype. However, a linear mixed effect model, followed with a posthoc test (Tukey Honest Significant Difference, HSD, test) comparing the difference between the means of Ctrl and KO at each time point showed a significant difference (at adjusted alpha 5%) for the time point T45 only, in line with the result we initially obtained. Nevertheless, we updated the figure and adjusted the text to include this precision. Furthermore, we ran similar 2-way ANOVA analysis on other pieces of data (all PTT; ipGTT ,ITT and GlyTT tests; indirect calorimetry measures) and modified figures and texts when needed.

“Figure 1B: This data set is blood glucose measurements from the same animals taken at three different time points. Thus, the statistical test to use is again a repeated two-way ANOVA. The authors indicate in the legend that 4 animals were used for this experiment. However, data for n=12-13 are shown in the source data file. Moreover, when the data are analyzed (in SPSS) there are no time x genotype interaction in the data set and no genotype effect. Thus, it is not correct to add the two stars at T6 as done by the authors in the present version of the manuscript.”

As the reviewer mentioned, data were here obtained from measurements on the same mice at 3 different time points. Our goal here was to compare glycemia between genotypes at each time point. To better take this into consideration, we adjusted our t-test results for multiple testing using a Bonferroni correction. We updated the figure and performed a similar analysis for the Fig. 1c with the same experimental design. All results are in the source data file and text and figures were updated.

This said, we thank the referee for duly pointing out a discrepancy between the figure legend and the real numbers in the data source file. This was a mistake from our side. The correct numbers were the ones in the data source file, and have now been corrected in the figure legend. We have corrected also a few additional similar mismatches in other figures.

“Figure 1F: Here I only tested the C1+C11 data set from fed and fasted WT and KO animals. The test to use is a normal two-way ANOVA. As indicated below (the output is from SigmaPlot) there is a main effect of genotype and no other statistical significant effects. Thus, the authors cannot differentiate between the fed and fasted condition as they have done now. The correct way to include the statistical indicators would be to add three stars above both black bars. Finally, the authors indicate in the legend that n=4 animals were used for these experiment. However, they include data from 5-6 animals in the source data file.”

In this data set, we compared fed and fasted states between 2 genotypes, so measurements were not repeated. As suggested by the reviewer, we performed a classical 2-way ANOVA and found again a significant effect of the genotype even after correction for multiple testing. We modified the figure accordingly, yet this change, again, does not affect our conclusions.

“Figure 5A: Again, a straight-forward two-way ANOVA should be applied. The output (from SigmaPlot) shows a main effect of genotype. Thus, one star above the black bars should be included in the figure, and the statistical indicator in the present version of the figure is not correctly placed. The interpretation of the data (lines 228-229) is clearly not supported by the statistics and the text should be revised.”

In this case, we agree with the reviewer than a 2-way ANOVA analysis showed an effect of the genotype on the NAD⁺ content. However, the aim of this analysis was to establish on which diet the genotype has a significant impact on NAD⁺ levels and thus the ANOVA test does not suit our biological question. Hence, this is the reason why we performed t-tests on each diet. Therefore, we believe that our conclusion is accurate to the result of our interrogation.

Referee 5:

“Endogenous Nicotinamide Riboside metabolism protects against diet-induced liver damage seems to be a well-written article.

But, I am a statistician and this was the reason I was consulted. Since I have little knowledge of the actual subject of the article, I will only refer to the statistical aspects of the article in my review.

The “Statistical analyses” section of the article is rather brief, but this is also due to the fact that the statistical methods used in the analysis of this paper are purely standard procedures, appropriate to the small number of cases in the samples. The sample sizes are mostly 6-7, a maximum of 10 mice was used. This is standard in such experiments. However, this implies that such an analysis is purely explorative and has no confirmatory character, which means that hypothesis tests and especially p-values should not be overinterpreted. Due to the small number of cases, the power, i.e. the probability of rejecting a hypothesis if an effect is given, is very low. Hypothesis tests are always constructed in such a way that one should never infer the confirmation of the hypothesis from a non-significance, but in such a case this is particularly critical. In the present article, at one point or another, this is happening. For example on p.6, l.116 “... did not influence

skeletal muscle respiratory ...” or on p. 7, l. 146 “... was not due to alteration of the gluconeogenic gene expression...”.

One could discuss whether a non-parametric test, i.e. a Mann-Whitney-U-test, should be used instead of the t-tests due to the small number of cases. The article does not mention anything about the normal distribution assumption of the examined variables. However, due to the fact that I would only consider such an analysis as presented here to be explorative anyway, this is not strictly necessary. Additionally, the t-test is still quite robust even against violations against the normal distribution assumption.

In general, the article should focus more on effect sizes and standard deviations (also due to the explorative character already explained above), these are completely missing in the actual article and can only be read approximatively from the graphics.

Overall, a large number of hypothesis tests were performed. The authors were apparently aware that this could lead to a multiple test problem, but they only adjusted for multiple testing within individual settings. In view of the very large number of tests performed overall, one is nevertheless confronted here with a high type I error inflation. In my opinion, multiple testing can be dispensed with completely, in return it must be made clear that all p-values are to be understood purely descriptively. This also reflects the explorative character of the analysis. Again, this should be made clearer in the course of the article.”

We thank the referee for the independent assessment of our statistical methods. We fully agree with the conclusions. We have toned down our statements when the conclusions stemmed from a non-significant difference. We also made a clear statement in the methods section mentioning that given the exploratory character of the analyses, all p-values are to be understood purely descriptively.